**Increase of dissolved inorganic carbon and decrease of pH in near surface**
**waters of the Mediterranean Sea during the past two decades**
Liliane. Merlivat [a], Jacqueline. Boutin [a], David. Antoine [b,c], Laurence. Beaumont [d], Melek.
Golbol [b], Vincenzo. Vellucci. [b]
[a] Sorbonne Universités (UPMC, Univ Paris 06)-CNRS-IRD-MNHN, LOCEAN Laboratory,
F-75005 Paris, France
[b] Sorbonne Universités (UPMC, Univ Paris 06)-CNRS, LOV, Observatoire Océanologique,
Villefranche-sur-Mer 06230, France
[c] Remote Sensing and Satellite Research Group, Department of Physics and Astronomy,
Curtin University, Perth, WA 6845, Australia
[d] Division Technique INSU-CNRS, 92195 Meudon Cedex, France
Corresponding author: L. Merlivat (merlivat@locean.upmc.fr)
**Abstract**
Two three-year-long time series of hourly measurements of the fugacity of $CO_2$ ($fCO_2$) in the
upper 10m of the surface layer of the northwestern Mediterranean Sea have been recorded by
CARIOCA sensors almost two decades apart, in 1995-1997 and 2013-2015. By combining
them with alkalinity derived from measured temperature and salinity, we calculate changes of
pH and dissolved inorganic carbon (DIC). DIC increased in surface seawater by $\sim 25$ μmol
kg$^{-1}$ and $fCO_2$ by 40 μatm, whereas seawater pH decreased by $\sim 0.04$ (0.0022 yr$^{-1}$). The DIC
increase is about 15% larger than expected from equilibrium with atmospheric $CO_2$. This
could result from natural variability, e.g. the increase between the two periods in the
frequency and intensity of winter convection events. Likewise, it could be the signature of the
contribution of the Atlantic Ocean as a source of anthropogenic carbon to the Mediterranean
Sea through the strait of Gibraltar. Under this assumption, we estimate that the part of DIC
accumulated over the last 18 years represents ~30% of the total change of anthropogenic
carbon since the beginning of the industrial period.

**1 Introduction**
The concentration of atmospheric carbon dioxide ($CO_2$) has been increasing rapidly over
the 20[th] century and, as a result, the concentration of dissolved inorganic carbon (DIC) in
the near surface ocean increases, which drives a decrease in pH in order to maintain a
chemical equilibrium. These changes have complex direct and indirect impacts on
marine organisms and ecosystems [*Gattuso and Hansson*, 2011]. Empirical methods to
estimate the anthropogenic $CO_2$ penetration in the ocean since the industrial revolution
have improved over the past few decades [*Chen and Millero*, 1979; *Gruber et al.*, 1996];
[*Sabine et al.*, 2008]; [*F Touratier and Goyet*, 2004; 2009; *Woosley et al.*, 2016]. As the
concentration of anthropogenic carbon, $C_{ant}$, cannot be distinguished from the natural
background of DIC through total DIC measurements, these methods are based on the
analysis of different chemical properties of the water column. Direct estimates of the
anthropogenic $CO_2$ absorption in the sea surface layers are difficult owing to the large
natural variability driven by physical and biological phenomena. [*Bates et al.*, 2014] have
extracted the trend from the large variability, based on analysis of a long time series
(monthly or seasonal sampling). For the global surface ocean, [*Lauvset et al.*, 2015] have
used the Surface Ocean $CO_2$ Atlas (SOCAT) database [*Bakker et al.*, 2014] combined with
an interpolation method. Estimates of anthropogenic storage in the Mediterranean Sea
differ by about a factor of two [*Huertas et al.*, 2009; *F Touratier and Goyet*, 2009]. In
addition to the anthropogenic signal, oceanic DIC can also be the signature of a strong
interannual variability. In the North Atlantic, for instance, McKinley et al. [2011] has
shown that the long term trend emerges after more than 25 years because of natural
variability.
A high frequency sampling of the seawater carbon chemistry at the air-water interface over
extended periods of time is useful to assess trends and variability of DIC. In this paper we
analyze two three-year time series of hourly fugacity of $CO_2$, $fCO_2$, measured with
autonomous CARIOCA sensors [*Copin-Montégut et al.*, 2004; *Merlivat and Brault*, 1995] in
1995-1997 and 2013-2015, at two nearby locations in the northwestern Mediterranean Sea
(Fig. 1). Using measured $fCO_2$, temperature (T) and salinity (S), we derive the other variables
of the carbonate system (pH and DIC). The experimental setting is first described, and the
recent data obtained over the 2013-2015 period are presented. Combined with the 1995-1997
measurements previously published [*Hood and Merlivat*, 2001], we estimate the decrease of
pH and the increase of DIC. The results are discussed with respect to the contributions of the
exchange with atmospheric $CO_2$, to the possible impact of vertical mixing and to recent
estimates of the transport of anthropogenic carbon from the Atlantic Ocean over a 18 years
period.

**2 Material and methods**
**2.1**-The BOUSSOLE and DYFAMED sites

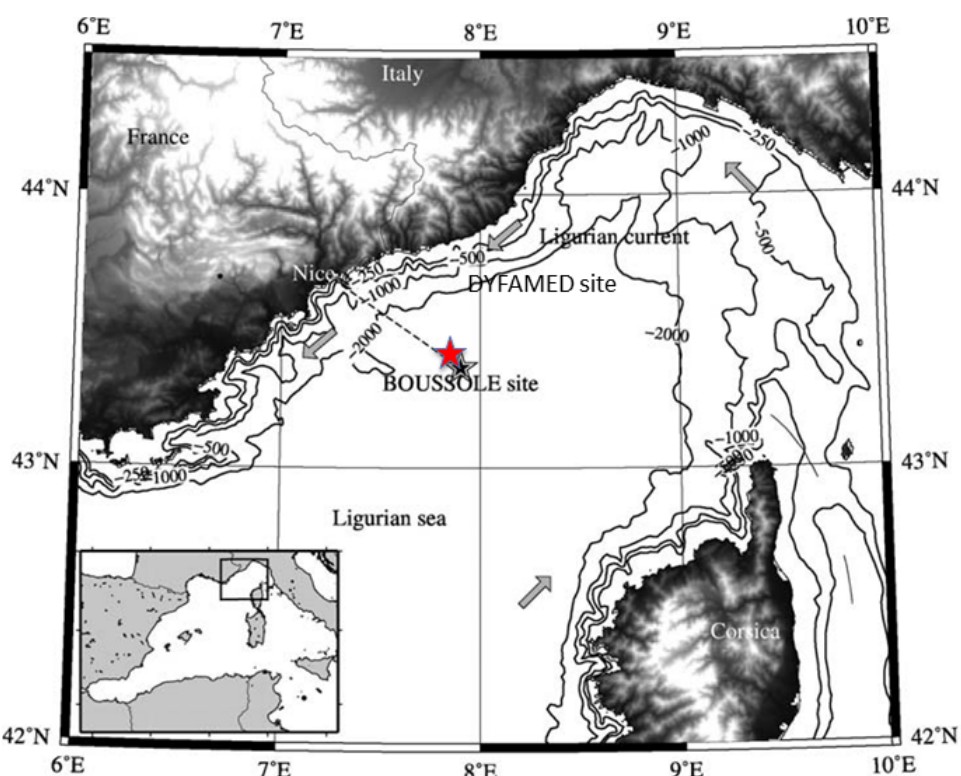


Fig.1. The area of the northwestern Mediterranean Sea showing the southern coast of France,
the Island of Corsica, the main current branches (gray arrows), and the location of the
DYFAMED site (43°25'N, 7°52'E, red star) and the BOUSSOLE buoy (43°22'N, 7°54'E,
black star) in the Ligurian Sea.

Data collection was carried out at the BOUSSOLE site (43°22'N, 7°54'E) in 2013-2015
[*Antoine et al.*, 2008; *Antoine. and others*, 2006] and at the DYFAMED site (43°25'N,
7°52'E) in 1995-1997 [*J.C. Marty et al.*, 2002]. These sites are 3 nautical miles apart, both
located in the Ligurian Sea, one of the basins of the northwestern Mediterranean Sea (Fig.1).
The water depth is of ~2400 m. The prevailing ocean currents are usually weak (<20 cm s$^{-1}$),
because these sites are in the central area of the cyclonic circulation that characterizes the
Ligurian Sea. The two sites surrounded by the permanent geostrophic Ligurian frontal jet
flow are protected from coastal inputs [*Antoine et al.*, 2008; *Heimbürger et al.*, 2013; *Millot*,
1999]. Monthly cruises are carried out at the same location .

**2.2**- Analytical methods
At DYFAMED, $fCO_2$ measurements at 2 m were provided by an anchored floating buoy
fitted with a CARIOCA sensor. At BOUSSOLE, measurements were carried out from a
mooring normally dedicated to radiometry and optical measurements, and onto which two
CARIOCA sensors were installed. Both monitored $fCO_2$ hourly at 3 and 10 m depth (although
only one of the two depths was equipped with a functional sensor at some periods); S and T
were monitored at the same two depths using a Seabird SBE 37-SM MicroCat instrument.
The CARIOCA sensors were adapted to work under pressure in the water column. They were
swapped about every 6 months, with serviced and calibrated instruments replacing those
having been previously deployed. The accuracy of CARIOCA $fCO_2$ measurements by the
spectrophotometric method based on the optical absorbance of a solution thymol blue diluted
in seawater is estimated at 2 µatm during both periods. Hood and Merlivat [2001] have
reported agreement between $fCO_2$ measured by CARIOCA buoys, similar to the one deployed
at DYFAMED, with ship based measurements, during a number of field programs, with an
accuracy of 2 µatm and a precision of 5 µatm .
At Boussole, newly designed $fCO_2$ sensors have been calibrated using in situ seawater
samples taken at 5 and 10 m depth during the monthly servicing cruises to the mooring. The
total alkalinity, Alk, and DIC of the samples were determined by potentiometric titration
using a closed cell according to the method developed by [*Edmond*, 1970]. Certified
Reference Materials (CRMs) supplied by Dr. A.G. Dickson (Scripps Institution of
Oceanography, San Diego, USA) were used for calibration [*Dickson et al.*, 2007]. The
accuracy is estimated at 3 µmol kg$^{-1}$ for both DIC and Alk. $fCO_2$ is calculated using the
dissociation constants of Mehrbach refitted by Dickson and Millero [*Dickson and Millero*,
1987; *Mehrbach et al.*, 1973] as recommended by Alvarez et al.[2014] for the Mediterranean
Sea. Error on $fCO_2$ derived from an individual sample is expected to be on the order of 5
µatm [*Millero*, 2007]. About 8 samples have been used to calibrate each CARIOCA sensor so
that the error on the absolute calibration of each $fCO_2$ CARIOCA sensor, is estimated at 1.8
µatm. In addition, we observe that the standard deviation of the difference between the
CARIOCA $fCO_2$ and $fCO_2$ computed with the monthly discrete samples (Fig. 2b) is equal to
4.4 μatm, consistent with the expected precision on CARIOCA $fCO_2$ of 5 μatm. Alk and S of
the 56 samples taken at BOUSSOLE are linearly correlated according the following
relationship :
$$\text{Alk (μmol kg}^{-1}) = 87.647\ S - 785.5 \qquad (1)$$
The standard deviation of the Alk data around the regression line is equal to 4.4 μmol kg$^{-1}$
($r^2 = 0.89$).

**3 Results**
**3.1** The BOUSSOLE mooring (2013-2015) time series
Temperature and $fCO_2$ were measured from February 2013 to February 2016. All seasons
were well represented, with missing data only in May-July 2013. For some periods,
simultaneous measurements were made at 3 and 10 m depth (Fig. 2, a, b, c).

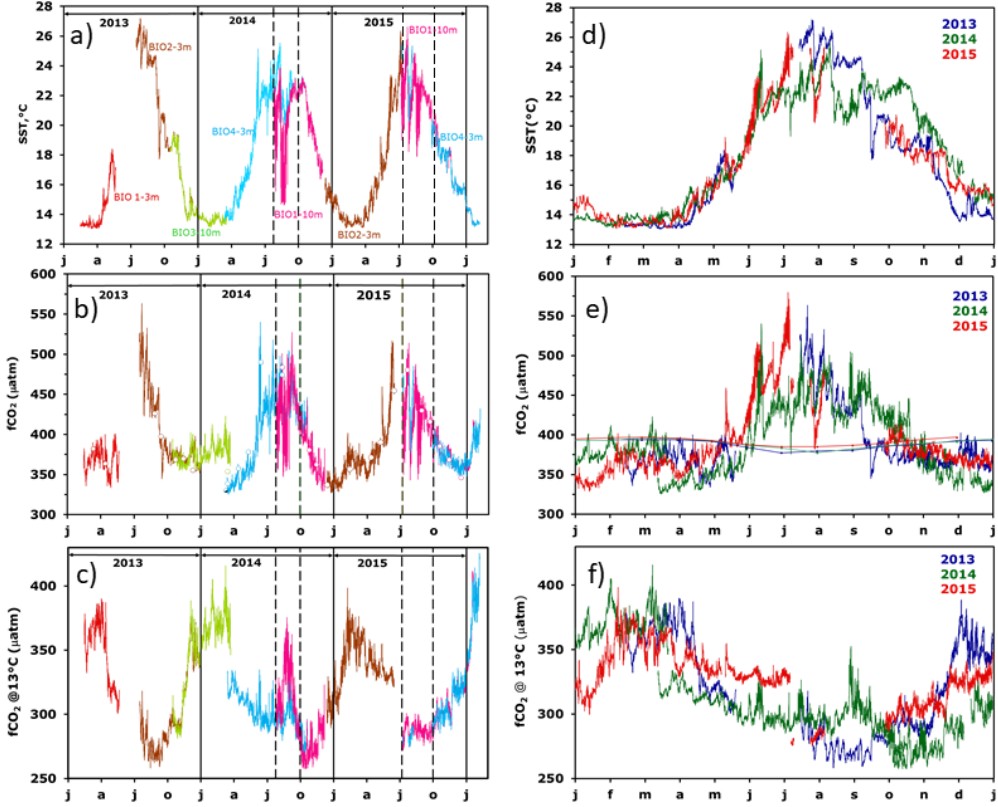


Fig.2. Interannual variability of CARIOCA data: a) T, b) $fCO_2$, c) $fCO_2@13$. The dotted lines
indicate the period affected by stratification and internal waves (July, 26 $^{th}$ to October 1$^{st}$,
2014 and July, 8 $^{th}$ to October 1$^{st}$, 2015). On 2(b), the open circles correspond to fCO2 data
derived from DIC and alkalinity measurements of samples taken at 5 and 10 m. (d), (e), (f),
seasonal variability. On 2(e), the thin lines indicate $fCO_{2atm}$. Note that the color code on (d),
(e), (f) is different from (a), (b), (c).

The range of temperature (Fig. 2a) extends from 13°C in winter up to 27°C in summer,
followed by progressive cooling in fall. The coldest temperature, 13°C, results from the
winter vertical mixing with the deeper Levantine Intermediate Water, LIW, marked by
extrema in temperature and salinity [*Copin-Montegut and Begovic*, 2002]. Temperature
provides the main control of the seasonality of $fCO_2$, from 350 µatm to more than 550 µatm in
summer 2013 (Fig. 2b). The fugacity of $CO_2$ in seawater is a function of temperature, DIC,
alkalinity, salinity and dissolved nutrients. In the oligotrophic surface waters of the
Mediterranean Sea, the effect of nutrients may be neglected. Temperature and DIC have the
strongest influences. By normalizing $fCO_2$ to a constant temperature, the thermodynamic
effect can be removed and changes in $fCO_2$ resulting from changes in DIC can be more easily
identified. Figure 2c shows the variability of $fCO_2$ normalized to the constant temperature of
13°C, ($fCO_2$@13), using the equation of [*Takahashi et al.*, 1993]. The underlying processes
that govern the seasonal variability of $fCO_2$@13 are successively winter mixing, biological
activity (organic matter formation and remineralization) and deepening of mixed layer in fall
[*Begovic and Copin-Montegut*, 2002; *Hood and Merlivat*, 2001]. Biology accounts for the
decline in $fCO_2$@13 observed from March-April to late summer; the ensuing increase of
surface $fCO_2$@13 is associated with the deepening of the mixed layer in the fall or convection
in winter as the vertical distribution of $fCO_2$@13 at DYFAMED shows a maximum in the 50-
150 m layer where a large remineralization of organic matter occurs, the productive layer
being mostly between 0 and 40 m [*Copin-Montegut and Begovic*, 2002]. The contribution of
air-sea exchange is not significant [*Begovic and Copin-Montegut*, 2002]. Over the period
2013-2015, the $CO_2$ air-sea flux from the atmosphere to the ocean surface is equal to -0.45
mol $m^{-2}$ $yr^{-1}$.
During summer 2014, large differences between measurements at 3 and 10 m were
observed (Fig. 2, a, b, c between dashed lines). A detailed analysis of the temporal
variability during that period underscores the role of inertial waves at the frequency of
17.4 hours that create the observed differences between the 2 depths of observations,
the deeper waters being colder and enriched in $fCO_2$@13. T and $fCO_2$@13 variability is
dominated by inertial waves. In particular, from 15 to 26 of August 2014, the difference
in T between the two depths is as large as 7.6°C, and 5.1°C on average. $fCO_2$ decreases on
average by 32.7 µatm leading to an increase of $fCO_2@13$ equal to 42.8 µatm.
The 2013-2015 seasonal and inter-annual variability of T, $fCO_2$ and $fCO_2@13$ is
illustrated on Fig. 2, d, e, f. The larger interannual changes in temperature (Fig.2, d) are
observed during summer, both at 3 m and 10 m depth, while over February and March, a
constant value of 13°C is observed as the result of vertical mixing with the LIW. A very
large inter-annual variability of $fCO_2@13$ is observed for T<14°C (Fig. 2,f). This is
associated with the winter mixing at the mooring site, which is highly variable from year
to year. Winter mixed-layer depth, MLD, varies between 50 and 160 m, at the top of the
LIW over the 2013-2015 period [*Coppola et al.*, 2016]. The variable depth of the winter
vertical mixing causes the difference in $fCO_2@13$ as $fCO_2$ increases with depth [*Copin-*
*Montegut and Begovic*, 2002]. The deepening of MLD is driven by episodic and intense
mixing processes characterized by a succession of events lasting several days, related to
atmospheric forcing [*Antoine et al.*, 2008] which lead to increase in $fCO_2@13$. Figure 2,e
illustrates the solubility control of the variability of $fCO_2$, as $fCO2$ increases when T
increases. Another cause of inter-annual variability of $fCO_2$ for T~14°C is the timing of
the spring increase of biological activity which differs by a month between years; for
instance, it happened at the beginning of April in 2013, T~15-16°C and by mid March in
2014, T~14°C. Another cause is the deepening of the mixed layer due to the fall cooling
which varies by a month between years.

**3.2** Decadal changes of hydrography
**3.2.1** Sea surface temperature changes
Monthly mean values of temperature have been computed for the two three-year periods,
1995-1997 and 2013-2015. In 1995-1997, $fCO_2$ and T at 2 m were measured with CARIOCA
sensors installed on a buoy at DYFAMED [*Hood and Merlivat*, 2001]. The mean annual
temperature of hourly CARIOCA data is equal to 18.21°C. For 2013-2015, temperature
measurements made on the BOUSSOLE mooring at 3 and 10 meters have been used. For the
April to September time interval, there are only data at 3m depth. In addition, temperature
data measured half hourly at 0.7 m at a nearby meteorological buoy (43°23'N, 7°50'E)
(http://www.meteo.shom.fr/real-time/html/DYFAMED.html) have been used (Fig.3d). Mean
annual temperature are equal to 18.29°C and 17.97°C respectively, based on the
meteorological buoy and the BOUSSOLE mooring data. The two sets of data differ
essentially during July and August, with the temperatures at 3 m being colder than at 0.7 m,
indicating a thermal gradient between the two depths during summer. Therefore, for 2013-
2015, we select the mean annual value computed with the meteorological buoy, 18.29°C, as
better representing the sea surface. This value is very close to 18.21°C computed for 1995-
1997. Then, no significant change of SST is found between the 2 decades, with a mean value
equal to 18.25°C.
**3.2.2** Sea surface salinity changes
The mean value of salinity computed from 56 samples taken at BOUSSOLE in 2013-2015 is
equal to 38.19+/-0.14. In 1998-1999, ship measurements of surface salinity were made during
monthly cruises at the DYFAMED site [*Copin-Montégut et al.*, 2004]. The mean salinity of
this set of 19 data is equal to 38.21+/-0.12. Thus, there is no significant salinity change
between the two decades.

**3.3** Decadal changes of $fCO_2@13$
**3.3.1** Time series of $fCO_2@13$ in 1995-1997 and 2013-2015
The two time series of high frequency data were analyzed in order to quantify the change of
$fCO_2@13$ at the sea surface two decades apart. To account for the interannual seasonal
variability as well as irregular sampling, we performed an analysis of the change of $fCO_2@13$
as a function of SST (Fig. 3, a and b). For the 2013-2015 data set, we excluded summer data
measured at 10 m depth as they were not representative of the surface mixed layer due to a
strong stratification. Much larger $fCO_2@13$ values are observed at low temperature than at
high temperature, the decrease being similar for the two studied periods and strongly non
linear. As described in section 3.1, large values at low temperature result from mixing with
enriched deep waters during winter and low values for 26°C-28°C temperatures occur at the
end of summer after biological drawdown of carbon. An increase of $fCO_2@13$ between the 2
periods is clearly highlighted for the whole range of temperature.

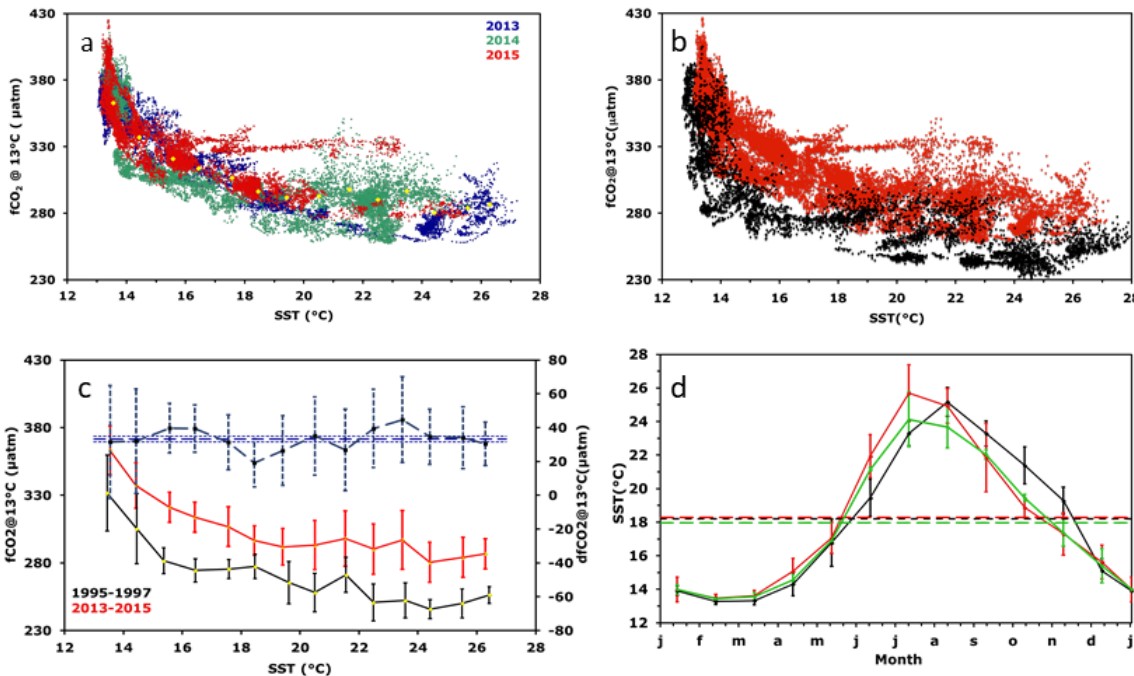


Fig.3. (a) fCO$_2$@13 as a function of temperature for hourly data in 2013, 2014 and 2015.The
yellow dots indicate mean fCO$_2$@13 (b) as in (a) but for all hourly data in 1995-1997 (black)
and in 2013-2015 (red) (c) As in (b), but for average values per 1°C interval (standard
deviation as dotted line). The difference between the two periods is also displayed (dashed
black curve; scale on the right axis). (d) Mean monthly sea surface temperature for 1993-1995
(black curve; CARIOCA sensors), 2013-2015 (green; CARIOCA sensors), 2013-2015 (red,
meteorological buoy). Corresponding mean annual values are indicated by dotted lines.

**3.3.2** Trend analysis and statistics
To quantify the change of fCO$_2$@13 between the two data sets, we proceed as follows: data
are binned by 1°C temperature intervals, thereby removing any potential seasonal weighting,
especially towards the 13-14°C winter months temperature. The measurements made in this
temperature interval represent about 25% of the total number of data for both periods. For
each of the fourteen 1°C step, the mean and standard deviation of hourly fCO$_2$@13
measurements are reported in Table 1 and on Fig. 3c.




| Time interval 1995-1997 | | | | Time interval 2013-2015 | | | | Temporal change | |
|---|---|---|---|---|---|---|---|---|---|
| T[1] °C | fCO2@13 µatm | N | standard deviation µatm | T[1] °C | fCO2@13 µatm | N | standard deviation µatm | dfCO2@13 µatm | standard deviation µatm |
| 13.45 | 331.58 | 1212 | 28.09 | 13.55 | 363.14 | 6869 | 18.07 | 31.56 | 33.40 |
| 14.45 | 305.28 | 495 | 26.02 | 14.43 | 337.16 | 3270 | 16.65 | 31.87 | 30.89 |
| 15.37 | 281.54 | 447 | 9.62 | 15.57 | 321.10 | 3112 | 11.09 | 39.56 | 14.68 |
| 16.44 | 274.43 | 182 | 8.53 | 16.42 | 313.79 | 1818 | 11.09 | 39.36 | 13.99 |
| 17.58 | 275.54 | 190 | 7.04 | 17.56 | 306.83 | 1528 | 14.65 | 31.29 | 16.25 |
| 18.47 | 277.34 | 300 | 9.04 | 18.45 | 296.57 | 2621 | 10.95 | 19.23 | 14.20 |
| 19.62 | 265.43 | 342 | 15.58 | 19.41 | 291.84 | 1406 | 13.45 | 26.40 | 20.59 |
| 20.50 | 258.08 | 529 | 14.15 | 20.50 | 293.16 | 1135 | 18.21 | 35.08 | 23.06 |
| 21.56 | 271.15 | 239 | 12.98 | 21.54 | 297.96 | 1200 | 20.41 | 26.82 | 24.19 |
| 22.49 | 250.75 | 742 | 13.66 | 22.49 | 290.27 | 2385 | 18.57 | 39.52 | 23.05 |
| 23.57 | 252.22 | 320 | 13.00 | 23.47 | 296.92 | 747 | 21.77 | 44.70 | 25.36 |
| 24.41 | 245.85 | 506 | 7.08 | 24.40 | 280.44 | 959 | 14.82 | 34.59 | 16.43 |
| 25.50 | 250.06 | 215 | 10.77 | 25.53 | 284.05 | 456 | 14.81 | 33.99 | 18.31 |
| 26.42 | 256.29 | 279 | 6.24 | 26.29 | 286.71 | 249 | 11.23 | 30.42 | 12.85 |


Table 1:
Distribution of temperature, fCO2@13, and increase dfCO2@13 data binned by 1°C
temperature interval for the 2 periods 1995-1997 and 2013-2015.
The mean temperature within each 1° step differ for the two periods as the distribution of
individual measurements are not identical.
For both data sets, a monotonic relationship between $fCO_2$@13 and T is observed with
correlation coefficients respectively equal to -0.861 and -0.857. The difference in fCO2@13
between the two periods, dfCO2@13, is derived in each temperature step, as the difference
between column 2 and 6 of Table 1. The variability of this difference is estimated as the
quadratic mean of the standard deviation in each time series. Both values are reported in
Table 1, column 9 and 10, and on Fig. 3c.
The distribution of $dfCO_2$@13 values around the mean seems random and indicates no
trend dependency with SST (Fig. 3c). This suggests that the processes which control the
seasonal variation of $fCO_2@13$ at the sea surface have not changed over the last two
decades.
We have estimated the uncertainties in the estimates of the difference $dfCO_2@13$ with 2
methods. Firstly, the arithmetic mean of $dfCO_2@13$ is equal to 33.17µatm, with a standard
deviation, SD, and standard error, SE, respectively equal to 6.29 µatm and 1.68 µatm. A 95%
confidence interval is thereby achieved within 1.96 SE, i.e 3.29 µatm. A second approach
consists of computing a weighted average of the mean of $dfCO_2@13$. In this case, mean
weighted value of $dfCO_2@13$ over the whole range of temperature is estimated, the weights
being equal to the variance of $dfCO_2@13$ in each temperature step. It is equal to 32.70 µatm.
The weighted SD, and the associated SE, of the 14 data points are respectively equal to 4.85
µatm and 1.30 µatm. A 95% confidence interval is achieved within 2.54 µatm. The difference
between the two mean $dfCO_2@13$ estimates is 0.47 µatm, well below SE. In the following,
we have chosen the former method which produces a more conservative estimate.

**3.4** Changes of seawater carbonate chemistry in surface waters
We estimated the DIC and pH changes related to the increase of $fCO_2@13$ measured at the
sea surface 18 years apart, assuming a mean salinity equal to 38.2, a mean alkalinity equal to
2562.3 µmol kg$^{-1}$ following equation (1), and a mean in situ temperature, T, equal to
18.25°C. The dissociation constants of Mehrbach refitted by Dickson and Millero [*Dickson*
*and Millero*, 1987; *Mehrbach et al.*, 1973] were used. pH is calculated on the seawater scale.
The error on $dfCO2@13$ ,+/-3.3µatm, has been propagated to compute the uncertainty on
dDIC and $dpH_{SWS}$. This makes the implicit assumption that there is no systematic error on
DIC and $pH_{SWS}$ derived from $fCO2@13$ between the two time periods; in particular, mean
temperature and salinity remain the same (section 3.2). This is further discussed in section
4.1.We compute an increase of DIC, dDIC, equal to 25.2+/-2.7 $\mu$ mol kg$^{-1}$ (1.40+/-0.15
$\mu$ mol kg$^{-1}$yr$^{-1}$) and the decrease of $pH_{SWS}$ , $dpH_{SWS}$ equal to  -0.0397+/-0.0042 $pH_{SWS}$ unit (-
0.0022+/-0.0002 $pH_{SWS}$ unit yr$^{-1}$) (Table 2).

| | d $fCO_2$* @ 13 µatm | d $fCO_2$* @ T µatm | d DIC* µmolkg$^{-1}$ | d $pH_{SWS}$ *** pH unit | d$fCO_2$@T annual µatm yr$^{-1}$ | d DIC annual µmolkg$^{-1}$yr$^{-1}$ | d $pH_{SWS}$ ***annual pH unit yr$^{-1}$ |
|---|---|---|---|---|---|---|---|
| sea surface | 33.2 +/-3.3 | 41.4 +/-4.1 | 25.2 +/-2.7 | -0.0397 +/-0.0042 | 2.30 +/-0.23 | 1.40 +/-0.15 | -0.0022 +/-0.0002 |
| atmosphere Lampedusa data | | 34.3 +/-2.3 | **20.8 +/-1.3 | | 1.91 +/-0.13 | 1.15 +/-0.07 | |
| d$fCO_2$@$T_{air}$/d$fCO_2$@$T_{sea}$ | | 0.83 +/-0.10 | 0.83 +/-0.09 | | | | |


Table 2
Seasonally detrended long term and annual trends of seawater carbonate chemistry and
atmosphere composition.
T, mean annual temperature equal to 18.25°C
*, change from 1995-1997 to 2013-2015.
**, dDIC $_{ant}$
*** d$pH_{SWS}$ computed at T

**3.5** Changes in atmospheric and seawater $fCO_2$
The increase of atmospheric $fCO_2$ from 1995-1997 to 2013-2015 was computed from
monthly atmospheric $xCO_2$ concentrations measured at the Lampedusa Island station (Italy)
(35°31'N, 12°37'E) (http://ds.data.jma.go.jp/gmd/wdcgg/) (see equation 3 in [*Hood and*
*Merlivat*, 2001]). Considering a mean annual in situ temperature equal to 18.25°C and an
atmospheric pressure of 1 atm, we derived a mean atmospheric $fCO_2$ equal to 355.3+/-0.8
µatm for 1995-1997 and 389.6+/-0.9 µatm for 2013-2015, that is an increase of 34.3+/-2.3
µatm (95% confidence interval) (Table 2). At this temperature, the change of $fCO_2$ at the sea
surface is 41.4+/-4.1 µatm. Thus the contribution of the increase in atmospheric $CO_2$ is
responsible for 84+/-5 % of the increase of $fCO_2$ measured in the surface waters. With the
same salinity and alkalinity as previously, the corresponding change in surface DIC, assuming
air-sea equilibrium, would be 20.8+/- 1.3  µmol kg$^{-1}$ (Table 2).

**4 Discussion**
**4.1** Time change of surface alkalinity
High frequency measurements of $fCO_2$ and temperature over 2 periods of 3 years, 2 decades
apart, have allowed the computation of an increase of DIC equal to 25.1+/-2.3 μmol kg$^{-1}$
assuming no change of alkalinity. In the range of salinity of the BOUSSOLE samples, 37.9 to
38.5, the alkalinity values computed with Eq (1) are larger than those predicted by the
relationship established for the DYFAMED site, with a mean difference equal to 10+/-2 μmol
kg$^{-1}$ [*Copin-Montegut and Begovic*, 2002]. In both cases alkalinity measurements were made
with a potentiometric method using certified reference material supplied by A.G. Dickson for
calibration. It is difficult to identify the cause for a possible change of alkalinity between the 2
periods, 18 years apart, while no salinity change has been observed. At a coastal site 50 km
away from DYFAMED, Kapsenberg et al. [2017] have measured an increase of alkalinity
unrelated to salinity over the period from 2007 to 2015. They attribute it to changes in
freshwater inputs from land. However, based on data from Coppola et al. [2016], alkalinity in
the upper 50m at  DYFAMED did not change significantly from 2007 through 2014 (3.204
μmol kg$^{-1}$, P=0.0794,   r$^{*2}$=0.08). Thus, we cannot conclude on whether the difference
observed at DYFAMED/BOUSSOLE between the two periods is real or an artifact of
measurement techniques. As a sensitivity test, we compute the expected changes of DIC and
pH from 1995-1997 to 2013-2015 for a mean alkalinity increase of 10 μmol kg$^{-1}$: we get
annual changes, dDIC=+0.46 μmol kg$^{-1}$ yr$^{-1}$ and dpH=-0.0001 pH unit yr$^{-1}$, which are well
below errors estimated in section 3.4. Hence, such a change in alkalinity does not
significantly affect the increase of DIC and the decrease of pH shown in Table 2.

**4.2** Drivers of the temporal change of DIC in surface waters
The increase in sea surface DIC from 1995-1997 to 2013-2015 is 25.2+/-2.7 μmol kg$^{-1}$ (Table
2). The expected contribution due to ocean uptake of anthropogenic $CO_2$ is 20.8+/-1.3 μmol
kg$^{-1}$. The difference between these two values is significant. In order to interpret this
difference, we examine potential changes that may result from interannual variability in local
physical and biological processes or anthropogenic carbon invasion from lateral advection of
Atlantic waters.
**4.2.1** Natural variability
Time series of mixed layer depth, MLD, show a strong variability in winter at interannual
scale. During the two periods, 1995-1997 and 2013-2015, the winter MLD never exceeded
220 m, whereas values over 300 m were observed in 1999 and especially in February and
March 2006 with values close to 2000 m [Coppola et al., 2016; Pasqueron de Fommervault et
al., 2015]. These episodes of strong and deep vertical mixing must have entrained DIC rich
LIW in the surface waters. This could be causing an increase in DIC between the 1995-1997
and 2013-2015 periods. Monthly surface samples collected at the Dyfamed time series station
between 1998 and 2013 indicate an increasing DIC trend of 1.35 μmol kg$^{-1}$ yr$^{-1}$. This value is
known with great uncertainty (r$^2$ = 0.05) because of the large seasonal variability displayed in
the monthly samples [*Gemayel et al.*, 2015]. Nevertheless, this value is closer to the trend we
calculated between the two periods, 1993-1995 and 2013-2015 (1.40 μmol kg$^{-1}$ yr$^{-1}$) than to
the trend inferred from the atmospheric increase (1.15 μmol kg$^{-1}$ yr$^{-1}$). On DYFAMED time
series, we find no evidence that the strong increase in MLD observed during winters 1999 and
especially 2006 resulted in a further increase in DIC.
The monthly cruises of the Dyfamed time-series study have also been analyzed in order to
investigate the hydrological changes and some biological consequences over the period 1995-
2007 [*J. C. Marty and Chiavérini*, 2010]. These authors show that extreme convective mixing
events such as recorded in 1999 and 2006 are responsible of large increases in nutrient
content in surface layers and conclude that the biological productivity is increasing especially
during the 2003-2006 period, which could lead to a larger consumption of carbon, i.e. a
decrease of DIC.
**4.2.2** Anthropogenic carbon exchange through the Strait of Gibraltar.
The concentration of oceanic anthropogenic carbon, $C_{ant}$, is not a directly measurable
quantity. To estimate it, several empirical methods have been developed. Flecha et al.[2012]
computed the anthropogenic carbon inventory in the Gulf of Cadiz. They used observations
made during a cruise in October 2008 throughout the oceanic area covered by the Gulf of
Cadiz and the Strait of Gibraltar to estimate $C_{ant}$ with 3 methods: $\Delta C^*$ [*Gruber et al.*, 1996]
,TrOCA [*F Touratier and Goyet*, 2004; *F. Touratier et al.*, 2007] , $\varphi C_T^0$ [*Vazquez-Rodriguez*
*et al.*, 2009]. In the 3 cases, their results indicate a net import of $C_{ant}$ from the Atlantic
towards the Mediterranean through Gibraltar.
Schneider et al. [2010], using the transit time distribution method applied to a dataset of a
Mediterranean cruise in 2001, estimated a net anthropogenic carbon flux across the Strait of
Gibraltar into the Mediterranean Sea of 3.5 Tg C yr$^{-1}$. Over the whole period from 1850 to
2001, this contribution of $C_{ant}$ represents almost 10% of the total $C_{ant}$ inventory of the
Mediterranean Sea. Accordingly, about 90% must have been taken directly by equilibrium
with atmospheric $CO_2$. Based on a high-resolution regional model, Palmieri et al. [2015]
computed the anthropogenic carbon storage in the Mediterranean basin. They concluded that
75% of the total storage of $C_{ant}$ in the whole basin comes from the atmosphere and 25% from
net transport from the Atlantic through the Strait of Gibraltar. The findings of these two
studies support our estimated change of DIC in excess of 17+/-10% over the direct
contribution of air-sea exchange suggesting that it could result from the anthropogenic carbon
input from the Atlantic Ocean towards the Mediterranean basin.
Huertas et al. [2009] and Schneider et al. [2010] report $DIC_{ant}$ surface concentrations
respectively equal to 65-70 µmol kg$^{-1}$ at the Strait of Gibraltar in the years 2005-2007 and
close to 65 µmol kg$^{-1}$ in the western basin in 2001. We extrapolate these figures to the year
2014, assuming a mean increase rate of DIC equal to 1.4 µmol kg$^{-1}$yr$^{-1}$ as previously
computed (Table 2). Taking into account the increase of $DIC_{ant}$ equal to 25.2 µmol kg$^{-1}$
between 1995-1997 and 2013-2015, we would estimate that the contribution of the change of
$DIC_{ant}$ over the last 18 years represents ~30% of the total change since the beginning of the
industrial period (t>~1800).
**4.3** Long term trends in surface DIC and pH
The annual changes of DIC and $pH_{SWS}$ calculated between 1995-1997 and 2013-2015 are
respectively equal to 1.40 +/-0.15 µmol kg$^{-1}$ and -0.0022+/-0.0002. At the DYFAMED site, at
10 m, Marcellin Yao et al. [2016] studied the time variability of pH over 1995-2011, based on
measurements of T, S, Alk and DIC sampled approximately once a month. They computed a
mean annual decrease of -0.003 ± 0.001 pH units on the seawater scale that is not
significantly different from our estimate. For the global surface ocean, Lauvset et al. [2015]
have reported a mean rate of decrease of pH, -0.0018+/-0.0004 for 1991-2011. This value is
also within the limits of uncertainty of the pH change computed in our study.
Bates et al. [2014] examined changes in surface seawater $CO_2$-carbonate chemistry at the
locations of seven ocean $CO_2$ time series that have been gathering sustained observations
from 15 to 30 years with monthly or seasonal sampling. Six stations are located in the
Atlantic and Pacific oceans in a latitudinal band between 10° N and 68°N. The range of
increasing and decreasing annual trends of DIC and pH extends from 0.93 +/-0.24 to 1.89 +/-
0.45 µmol kg$^{-1}$yr$^{-1}$ and -0.0014+/-0.0005 to -0.0026+/-0.0006 respectively. The Revelle factor
of surfaces waters vary from 9-10 in the low latitude to 12-15 in the subpolar time series sites,
with higher Revelle factor values reflecting reduced capacity to absorb atmospheric $CO_2$. The
data show that the increase of DIC is not only controlled by the buffer capacity of the water
but compounding effects of changes in physical factors as strengthening of winter mixing or
larger air-sea uptake, have also to be taken into account [*Olafson et al.*, 2010].
The increase of DIC computed at DYFAMED is rather in the upper range of values reported
at the other time series. A low Revelle factor, close to 10, characterizes the Mediterranean
Sea because of its warm and high-alkalinity waters. Moreover, as the result of a relatively
short deep water renewal time estimated to be 20-40 years in the western basin[*Schneider et*
*al.*, 2010] , the waters of the Mediterranean Sea have a relatively high absorption capacity to
absorb anthropogenic $CO_2$ from the atmosphere and transport it to depth.
The calculated decrease of pH in surface water at DYFAMED and in the global ocean are
quite similar, despite the higher alkalinity of the Mediterranean Sea. Thermodynamic
equilibrium calculations have highlighted the alkalinity effect on the Mediterranean
anthropogenic acidification [*Palmiéri et al.*, 2015]. Their results show that, notwithstanding a
higher total alkalinity, the average anthropogenic change in surface pH does not differ
significantly from the global average ocean.

**5 Conclusion**
High-frequency ocean fCO2 measurements made by CARIOCA sensors were sufficient to
estimate trends in fCO2, DIC and pH over a period of two decades, notwithstanding a
considerable short-time and natural seasonal variability of these properties at the sea surface.
We have estimated a large change of sea surface carbonate chemistry, an increase of DIC and
a decrease of pH. The computed increase of DIC is larger than the change expected from
chemical equilibrium with atmospheric $CO_2$.This could be the result of a strong interannual
variability of the winter mixing as observed between the two periods 1993-1995 and 2013-
2015. Likewise, our results support modeling work and analysis of vertical profiles
measurements that suggest that the Atlantic Ocean contributes as a source of anthropogenic
carbon towards the Mediterranean basin, close to 10% ([*Schneider et al.*, 2010] or 25%
[*Palmiéri et al.*, 2015].

*Data availability*: Time series data from Dyfamed (19951997) are available in the SOCAT v3
database. Boussole data (2013-2015) will be available in SOCAT v6.

**Acknowledgments**
Seawater samples were analyzed for DIC and Alk by the SNAPO-CO2 at LOCEAN in Paris.
The CO2Sys toolbox of [*Pierrot et al.*, 2006] has been used for the calculations of DIC and
pH. The adaptation of CARIOCA sensors to high pressure has been supported by the BIO-
optics and CARbon EXperiment (BIOCAREX) project, funded by the Agence Nationale de la
Recherche (ANR,Paris). We are grateful for helpful comments from Gilles Reverdin on the
manuscript. Many thanks to Laurent Coppola who kindly provided additional MLD data at
Dyfamed.

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

Table 1:

Distribution of temperature, $fCO_2@13$, and increase $dfCO_2@13$ data binned by 1°C
temperature interval for the 2 periods 1995-1997 and 2013-2015 .
The mean temperature within each 1° step differ for the two periods as the distribution of
individual measurements are not identical.


| Time interval 1995-1997 | | | | Time interval 2013-2015 | | | | Temporal change | |
|---|---|---|---|---|---|---|---|---|---|
| $T^1$ °C | fCO2@13 µatm | N | standard deviation µatm | $T^1$ °C | fCO2@13 µatm | N | standard deviation µatm | dfCO2@13 µatm | standard deviation µatm |
| 13.45 | 331.58 | 1212 | 28.09 | 13.55 | 363.14 | 6869 | 18.07 | 31.56 | 33.40 |
| 14.45 | 305.28 | 495 | 26.02 | 14.43 | 337.16 | 3270 | 16.65 | 31.87 | 30.89 |
| 15.37 | 281.54 | 447 | 9.62 | 15.57 | 321.10 | 3112 | 11.09 | 39.56 | 14.68 |
| 16.44 | 274.43 | 182 | 8.53 | 16.42 | 313.79 | 1818 | 11.09 | 39.36 | 13.99 |
| 17.58 | 275.54 | 190 | 7.04 | 17.56 | 306.83 | 1528 | 14.65 | 31.29 | 16.25 |
| 18.47 | 277.34 | 300 | 9.04 | 18.45 | 296.57 | 2621 | 10.95 | 19.23 | 14.20 |
| 19.62 | 265.43 | 342 | 15.58 | 19.41 | 291.84 | 1406 | 13.45 | 26.40 | 20.59 |
| 20.50 | 258.08 | 529 | 14.15 | 20.50 | 293.16 | 1135 | 18.21 | 35.08 | 23.06 |
| 21.56 | 271.15 | 239 | 12.98 | 21.54 | 297.96 | 1200 | 20.41 | 26.82 | 24.19 |
| 22.49 | 250.75 | 742 | 13.66 | 22.49 | 290.27 | 2385 | 18.57 | 39.52 | 23.05 |
| 23.57 | 252.22 | 320 | 13.00 | 23.47 | 296.92 | 747 | 21.77 | 44.70 | 25.36 |
| 24.41 | 245.85 | 506 | 7.08 | 24.40 | 280.44 | 959 | 14.82 | 34.59 | 16.43 |
| 25.50 | 250.06 | 215 | 10.77 | 25.53 | 284.05 | 456 | 14.81 | 33.99 | 18.31 |
| 26.42 | 256.29 | 279 | 6.24 | 26.29 | 286.71 | 249 | 11.23 | 30.42 | 12.85 |





 Table 2

Seasonally detrended long term and annual trends of seawater carbonate chemistry and
atmosphere composition.
T,mean annual temperature equal to 18.25°C
*, Change from 1995-1997 to 2013-2015.
**, dDIC $_{ant}$
***, dpH $_{SWS}$ computed at T



| | d fCO$_2$* @ 13 μatm | d fCO$_2$* @ T μatm | d DIC* μmolkg$^{-1}$ | d pH$_{SWS}$*** pH unit | dfCO$_2$@T annual μatm yr$^{-1}$ | d DIC annual μmolkg$^1$yr$^1$ | d pH$_{SWS}$*** annual pH unit yr$^1$ |
|---|---|---|---|---|---|---|---|
| sea surface | 33.2 +/-3.3 | 41.4 +/-4.1 | 25.2 +/-2.7 | -0.0397 +/-0.0042 | 2.30 +/-0.23 | 1.40 +/-0.15 | -0.0022 +/-0.0002 |
| atmosphere Lampedusa data | | 34.3 +/-2.3 | **20.8 +/-1.3 | | 1.91 +/-0.13 | 1.15 +/-0.07 | |
| dfCO$_2$@T$_{air}$/dfCO$_2$@T$_{sea}$ | | 0.83 +/-0.10 | 0.83 +/-0.09 | | | | |



Figure 1. The area of the northwestern Mediterranean Sea showing the southern coast of
France, the Island of Corsica, the main current branches (gray arrows), and the location of the
DYFAMED site (43°25'N, 7°52'E, red star) and the BOUSSOLE buoy (43°22'N, 7°54'E,
black star) in the Ligurian Sea.

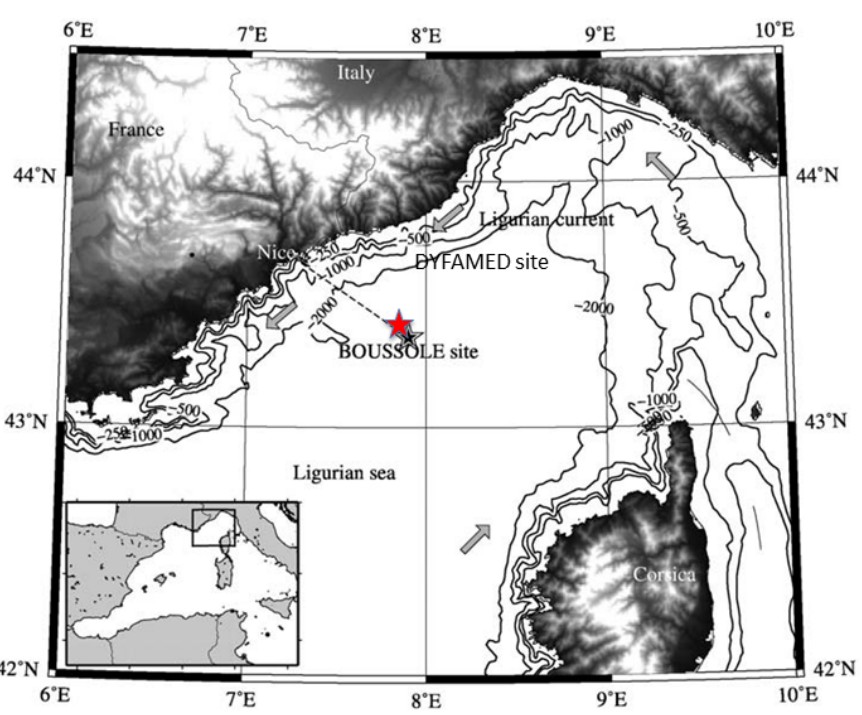



Figure 2. Interannual variability of CARIOCA data: a) T, b) $fCO_2$, c) $fCO_2@13$. The dotted
lines indicate the period affected by stratification and internal waves (July, 26[th] to October
1[st], 2014 and July, 8[th] to October 1[st], 2015). On 2(b), the open circles correspond to fCO2
data derived from DIC and alkalinity measurements of samples taken at 5 and 10 m. (d), (e),
(f), seasonal variability. On 2(e), the thin lines indicate $fCO_{2atm}$. Note that the color code on
(d), (e), (f) is different from (a), (b), (c).


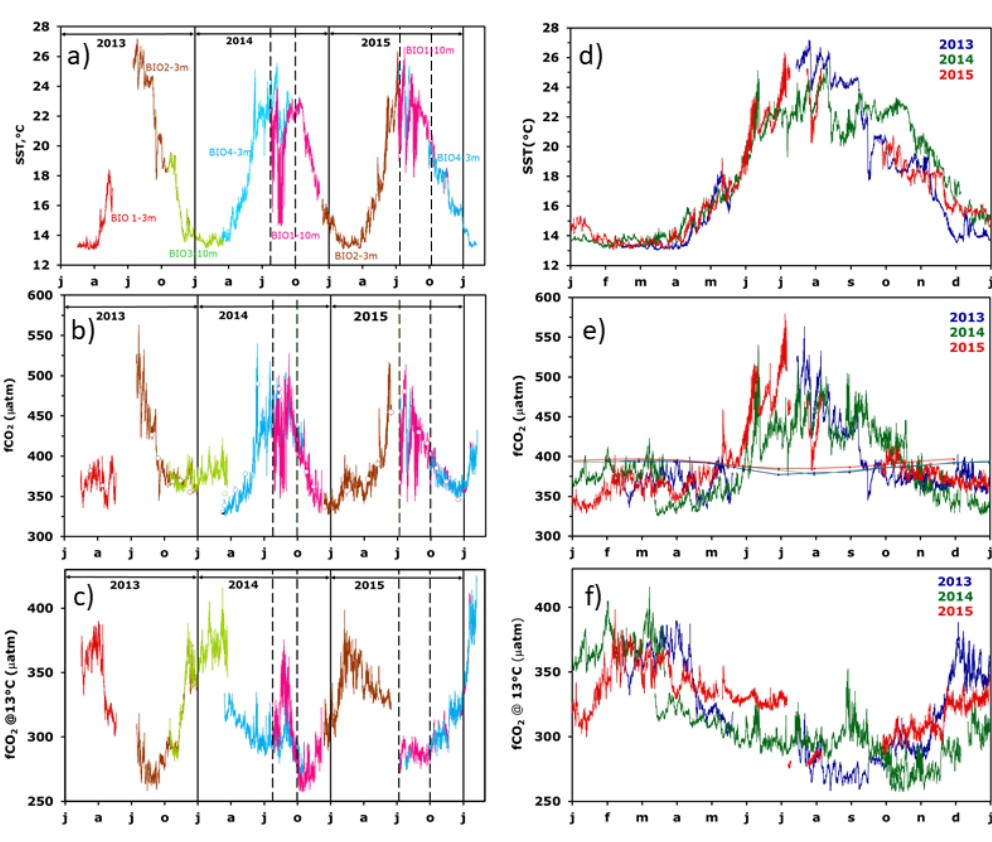



Figure 3. (a) fCO$_2$@13 as a function of temperature for hourly data in 2013, 2014 and
2015.The yellow dots indicate mean fCO$_2$@13 (b) as in (a) but for all hourly data in 1995-
1997 (black) and in 2013-2015 (red) (c) As in (b), but for average values per 1°C interval
(standard deviation as dotted line). The difference between the two periods is also displayed
(dashed black curve; scale on the right axis). (d) Mean monthly sea surface temperature for
1993-1995 (black curve; CARIOCA sensors), 2013-2015 (green; CARIOCA sensors), 2013-
2015 (red, meteorological buoy). Corresponding mean annual values are indicated by dotted
lines.

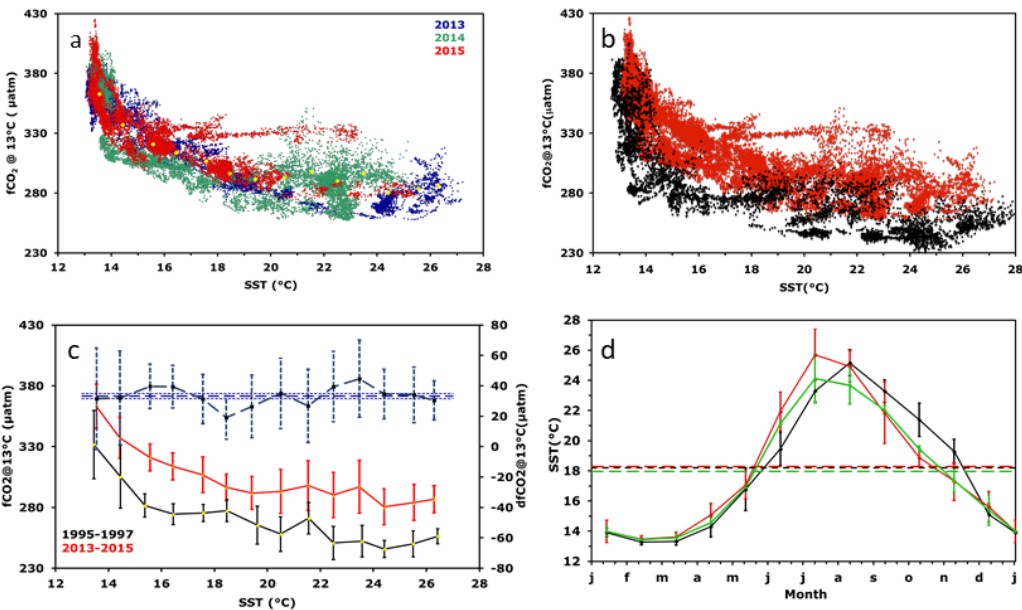
