# Peer review of "Increase of dissolved inorganic carbon and decrease of pH in near surface"

_Biogeosciences, 2017_

## Referee Comment (RC1) · Anonymous Referee #1 · 13 Aug 2017

The paper by Merlivat et al. provides a description of carbonate chemistry in two close fixed station located in the Ligurian Sea (northwestern Mediterranean Sea). By combining time series data of CO2 fugacity with alkalinity derived estimations, they reported an Increase of dissolved inorganic carbon and decrease of pH in near surface waters during the past two decades. This issue is of particular interest to the referee and I think that the authors have a very nice data set to exploit. However, I think the analysis is somewhat incomplete, and I finished the paper wanted a more in-depth analysis and discussion. I encourage the authors to further expand their work because at this stage their hypothesis are not well supported. The manuscript could be published in Biogeosciences after a major revision in order to clarify some aspects as indicated

below.

Major Comments

My major reservation about this work is the difference between the measured fCO2 at the sea surface (fCO2sea) and the fCO2 derived from atmospheric xCO2 concentration (fCO2air). In 2013-2015 the sea surface mean annual fCO2 calculated at 18.25°C (the mean annual in situ temperature) was larger than the fCO2air derived from atmospheric data at the same temperature. This result is quite strange, because it means a CO2 outgassing from the sea surface to the atmosphere on annual average, which is in contrast with respect to the ongoing ocean acidification process and the general net anthropogenic CO2 uptake measured in the Mediterranean Sea by different research. In 2013-2015 I would expect an equilibrium between the fCO2sea and fCO2air, or a slightly higher value in the fCO2air, as it was detected in the 1995-1997. How the authors can explain this issue? They suggested the contribution of the Atlantic Ocean as a source of anthropogenic carbon, but I do not understand how the Atlantic surface waters can be relatively enriched in anthropogenic carbon. Moreover, this is in contrast with the end of the discussion where the authors say that (P13L331) "The Mediterranean Sea is actually able to absorb more anthropogenic CO2 per unit area". Maybe there are other causes which could explain the fCO2 increase at the sea surface observed in 2013-2015, such as a stronger and deeper winter vertical mixing with CO2 enriched LIW. Finally, additional information about the water mass exchange throughout the Strait of Gibraltar and its temporal variation are needed. These can be found in the recent review of Jordà et al. (2017) which may provide more insights for this work.

The authors found a DIC increase larger than expected from equilibrium with atmospheric CO2. They hypnotized a ~15% contribution of the Atlantic Ocean as a source of anthropogenic carbon to the Mediterranean Sea through the strait of Gibraltar. I think that the analysis presented in the manuscript are not sufficient to support such hypothesis and the authors should provide a lot more analysis and discussions. Moreover, the Mediterranean Sea overturning circulation and the sites of dense water formation

could play a very important role in the sequestration of anthropogenic CO2 and in the ocean acidification of the Mediterranean Sea. I think that the authors should read the recent papers of Touratier et al. (2016), Ingrosso et al. (2017), and Krasakopoulou et al. (2017), who estimated the anthropogenic CO2 in the Gulf of Lion, Adriatic Sea, and the Aegean Sea respectively.

The authors try to assess the influence of physical and biological process on the seasonal and inter-annual variation of fCO2. To do this, they used a simple analysis of the change of fCO2@13 (fCO2 normalized to the constant temperature of 13°C) as a function of SST, which is not sufficient to achieve the scope. I suggest to quantify (1) the air-sea CO2 exchange and (2) the thermal/not-thermal contributions on the fCO2 variation with the method of Takahashi et al. (2002). In this way the authors could clarify how fCO2 seasonal variation is affected by physical (i.e. temperature, mixing, and air-sea CO2 exchange) and biological processes (i.e. photosynthesis, respiration, and calcification).

Specific Comments

P4L93: If the authors followed the standard operational procedures, the reference of Dickson et al. (2007) could be added to Edmond (1970).

P5L126: I propose to consider here the the method of Takahashi et al. (2002) and to present the temporal variation of the thermal and not-thermal fCO2 as differences (dfCO2) with respect to the February, chosen as reference month because it usually presents the lowest temperature and the minimum biological activity.

P5L128: The "remineralization" is a biological activity. Please modify/clarify the sentence.

P5L130: Do the authors have oxygen data? The examination of the O2/DIC or AOU (apparent oxygen utilization)/DIC ratio would provide useful information about the influence of biological activity to the observed fCO2 variation. Also satellite data of Chlorophyll a concentration may help, which nowadays are easy to get.

P6L134: "The contribution of air-sea exchange is not significant". In order to support this sentence, please can the authors calculate the air-sea $CO_2$ flux and estimate the real influence of this process?

P6L150: Levantine Intermediate Water (LIW) originates in the Eastern Mediterranean and takes years to reach the Ligurian Sea. Due to the organic matter remineralization processes, the LIW presents low dissolved oxygen concentration and high $CO_2$ levels (Álvarez et al., 2014), even higher than then the atmospheric levels. Taking into account these considerations, in the present study, the increase of total dissolved inorganic carbon observed in 2013-2015 can be related to a stronger and deeper winter vertical mixing with $CO_2$ enriched LIW? As reported by Alvarez et al. (2014), the LIW during its westward flows can increase DIC and lower pHT of different Mediterranean basin.

P7L197: "mixing with enriched deep waters" please substitute with "mixing with $CO_2$-enriched deep waters". This may support the hypothesis of a general DIC increase generated by mixing with LIW, but further analysis and more discussions are needed.

P8L199: During summer, due to the high sea surface temperature, the $CO_2$ flux from the sea to the atmosphere could also play an important role. Please consider also this process in addition to the biological drawdown of carbon.

P9L223: "Changes of seawater carbonate chemistry in surface waters". This section needs some modification/clarification. L223-227 seems more appropriate for the Material and methods. L229-234: DIC and pH are derived parameters. They are calculated from total alkalinity and $fCO_2$. Due to this reason, the $fCO_2$-DIC and $fCO_2$-pH may not have sense and the near perfect $R^2$ is not significant. Please, can the authors clarify this issue?

P9L229: pHT refers to the pH on the total scale. But the authors calculated the pH on the seawater scale (P9L228) which is conventionally denoted as pHsws. Please

substitute in all the manuscript/figures the pHT with pHsws.

P11L259: Any references which can support that Atlantic surface waters are relatively enriched in anthropogenic carbon and why? Even if the Atlantic surface water could be enriched in CO2, I do not think that it could preserve this property. An air-sea equilibrium, mixing, and biological processes may happen during the long time that Atlantic surface water spent to reach the Ligurian Sea from the Gibraltar Strait.

P11L270-272: More discussion and references are needed to support this sentence.

P13L335: More appropriate and recent references are Touratier et al. (2016), Ingrosso et al. (2017), and Krasakopoulou et al. (2017), who estimated the anthropogenic CO2 in the three dense water formation area of the Mediterranean Sea.

Technical comments

I suggest to improve the general quality of the figures.

P11L286: "P=0,0749" Substitute the coma with point.

References

Álvarez, M., Sanleón-Bartolomé, H., Tanhua, T., Mintrop, L., Luchetta, A., Cantoni, C., Schroeder, K., Civitarese, G., 2014. The CO2 system in the Mediterranean Sea: a basin wide perspective. Ocean Sci. 10, 69–92.

Carter, B. R., N. L. Williams, A. R. Gray, and R. A. Feely. 2016. Locally interpolated alkalinity regression for global alkalinity estimation. Limnol. Oceanogr. Methods 14: 268–277. doi:10.1002/lom3.10087

Dickson, A.G., Sabine, C.L., Christian, J.R., 2007. Guide to best practices for ocean CO2 measurements. PICES Spec. Publ. 3, 191.

Ingrosso, G., Bensi, M., Cardin, V., Giani, M., 2017. Anthropogenic CO2 in a dense water formation area of the Mediterranean Sea. Deep-Sea Research Part I 123, 118–

128. doi:10.1016/j.dsr.2017.04.004

Jordà, G., Schuckmann, Von, K., Josey, S.A., Caniaux, G., García-Lafuente, J., Sammartino, S., Ozsoy, E., Polcher, J., Notarstefano, G., Poulain, P.M., Adloff, F., Salat, J., Naranjo, C., Schroeder, K., Chiggiato, J., Sannino, G., Macías, D., 2017. The Mediterranean Sea heat and mass budgets: Estimates, uncertainties and perspectives. Progress in Oceanography 156, 174–208. doi:10.1016/j.pocean.2017.07.001

Krasakopoulou, E., Souvermezoglou, E., Giannoudi, L., Goyet, C., 2017. Carbonate system parameters and anthropogenic CO2 in the North Aegean Sea during October 2013. Continental Shelf Research 1–13. doi:10.1016/j.csr.2017.04.002

Touratier, F., Goyet, C., Houpert, L., de Madron, X.D., Lefèvre, D., Stabholz, M., Guglielmi, V., 2016. Deep-Sea Research I. Deep-Sea Research Part I 113, 33–48. doi:10.1016/j.dsr.2016.04.003

---

## Referee Comment (RC2) · Anonymous Referee #2 · 22 Sep 2017

In this manuscript, Merlivat et al. report on measurements of fCO2 during two 3-year windows whose midpoints are 18 years apart with samples taken adjacent locations in the Mediterranean Sea. They then combine those measurements with total alkalinity derived from measured temperature and salinity to compute DIC and pH. Because their derived DIC increase is larger than expected from equilibrium with atmospheric CO2, the authors invoke lateral transport of anthropogenic DIC from the Atlantic to the Mediterranean Sea to explain the difference.

GENERAL COMMENTS

The authors report on quality measurements of fCO2, the fruit of decades of investe-

ment to develop and deploy the CARIOCA buoys with fCO2 sensors. They use the same measurement system for all measurements, thus allowing an assessement of the total change in ocean fCO2 between the 2 time periods that seems as precise as can be hoped for. Yet despite the quality of the measurements, my impression is that the uncertainties are underestimated when the authors discuss temporal changes in measured fCO2 as well as derived DIC and pH. This impression comes partly from the authors' choice to represent uncertainties as the standard error of the mean rather than the standard deviation. Their estimated uncertainties for the difference between these two time periods is usually much smaller than the best measurement precision. For more about my concerns on the uncertainty analysis of the authors, please see the detailed comments below, e.g., those labeled line 214, lines 243-248, line 296, and line 320.

An even greater concern is that the authors assume that the total temporal change is entirely anthropogenically driven. They do not consider the potential contribution from natural variability (see detailed comments below for the section commenting on 'lines 44-46:'). Because of these two concerns, it appears to me that the manuscript may well require in-depth revisions before it is acceptable for publication.

DETAILED COMMENTS

lines 44-46: This statement from the authors in the introduction is an important one, making the point that there is large natural variability. Why then do they neglect to consider that natural decadal scale variabilty may explain part of the change between 1995-1997 and 2013-2017. In the North Atlantic, for instance it has been shown that because of decadal variability it requires 25 years for the long-term trend to emerge (McKinley et al, 2011). In the North Pacific, about half of the change in near surface ocean pH over a 15-year period has been ascribed to natural (non-anthropogenic) contributions (Byrne et al., 2010). In the Southern Ocean, early studies suggested a weaking of the Southern Ocean CO2 uptake, but more recent work with 30-year perspective indicates a tendency in the opposite direction, with such oscillations being ascribed in

part to natural variability (Lanschutzer, 2015). In contrast to these studies, the authors do not consider any contribution of natural decadal variability in their interpretation, assigning the measured and estimated changes entirely to an anthrogenically forced trend. The change between the 2 points in time, even if they represent 3-year averages as in this study by Merlivat et al., are also likely to be affected by natural variability.

lines 53-55: - please add "over extended periods" after air-water interface - please delete "related to the absorption of increasing atmospheric CO2 concentration" or nuance the message so as not to neglect natural variability.

lines 58-59: - please delete the commas just after "temperature" and just after "salinity" as these confuse the listing, making it appear longer that it is. You may also want add parentheses around 'T' and 'S', although I don't think that is necessary.

lines 76-77:

- Can you provide references to support your statement that the Ligurian current isolates the two stations from coastal inputs. I would expect that eddies and jets would allow some transfer of heat, salt, momemtum, and chemical species from coastal waters to the open Mediterranean Sea, even if that transfer is not occurring immediately adjacent to the two sampling sites.

- You could strengthen your case that the 2 stations (BOUSSOLE and DYFAMED) sample the same water mass by showing carbonate system measurements as well as T and S taken at the same time at both stations.

line 83: change "They" to "Both"

lines 96-98: - add "K1 and K2" before "dissociation constants" - Why do the authors choose to use the K1 and K2 from Dickson and Millero (1987) even though the first author of the paper, when asked, suggests that there is a mistake in those formulations? I think it would be better to use K1 and K2 from Lueker et al. (2000), which is recommended for best practices (Dickson et al., 2007).

[Figure]

line 106: This sentence could be ambiguous. Are you referring to the standard deviation of the all 56 samples? Please clarify.

line 110: The authors use the term "fCO2@13" before it has been defined. Would it not be simpler just to delete "and fCO2@13" and get to the details later.

lines 120-121: The fCO2 is also a function of total dissolved inorganic phosphorus and silicon, when computed from DIC and total alkalinity, although in the oligotrophic surface waters of the Med Sea those nutrient concentrations are negligible and do not contribute significantly.

line 123: Did Takahashi et al. (1993) study the Med Sea? If not, how do you make the connection.

line 130: change "decay of" to "decline in"

line 131: You could improve sentence flow by adding add "the ensuing" before "increase.

lines 134-135: The authors should provide evidence for their statement that the contribution of the air-sea flux is insignificant.

line 140: change "15th to 26th" to "15 to 26".

line 142: The meaning of "Likewise" is not clear. Please modify sentence to clarify your meaning.

line 201: The word "monotonous" means "boring" in English, perhaps not what was intended. I would suggest to use "monotonic" instead.

line 214: - By "standard error" I presume that the authors are using the 'standard error of the mean', the latter 3 words which should be added to make it clearer to readers. - I have several problems with the authors' choice to use the standard error of the mean (SE) in this case.

* First it gives the wrong impression that the uncertainty of these calculations is small (1.7 $\mu$atm), even lower than the precision of individual fCO2 measurements (3 $\mu$atm). Because the SE is the standard deviation divided by the square root of N, it is nearly 5 times smaller than the standard deviation in this case (N=24, Table 1).

* Second, the result for the SE will also depend on the authors' arbitrary choice for the scale.

* Third, even if the SE were appropriate, I do not understand how the authors get N=24 for the 'daily scale' mentioned in Table 1.

* Fourth, The use of SE in the right hand portion of Table 1 is at least visually inconsistent with the use of the standard deviation for each of the time periods shown in the left and center portions of the same table. I would stongly recommend that the authors simply use the standard deviation at least in Table 1. If the authors insist on using SE, I would ask that they also provide the standard deviation in addition to the SE and that they statistically justify the use of the SE while explaining their choices in detail (e.g., N=24). There have been comments in scientific journals about the misuse of SE being a common practice. The SE could perhaps be used correctly here if well justified, but it can also mislead readers.

line 215: The text says that "fCO2@13 is evenly distributed *in* the whole range of temperature". I am not sure I understand. It is seen in Table 1 that fCO2@13 varies from 19 to 45. Please clarify this sentence.

line 217: Change "2 last decades" to "last two decades".

lines 228: You say that pH is on the seawater scale but later you use pHT, meaning it is on the total scale. Please clarify.

line 231: The text says, "We used these sensitivity factors to compute the increase in DIC, ..." It is not clear why you need these sensitivity factors. Can you not simply compute DIC and pH for both time periods then take the difference?

line 232: The numbers for the increase in DIC are given with too many significant figures.

Table 2: The numbers for dfCO2 and dDIC are given with 4 significant figures, much too much. The number of significant figures gien in the paper is often too many. The authors should carefully go over the reported numbers and reduce to a justifiable number of significant figures in every case.

lines 243-248: - Please inform the reader what the error bars are reporting, standard deviation or standard error of the mean. - There is insufficient information about how 'atmospheric fCO2' was calculated from atmospheric xCO2. Did the authors make a humidity correction, which can change numbers by a few percent? Nothing along those lines was mentioned. How much of a difference would there be if the authors did not assume that the atmospheric pressure is 1 atm. Did they make the xCO2-to-fCO2 conversion on a monthly basis and then take an annual average? Currently it seems they are making only an annual-mean calculation. Would results differ? - The error esimate appears to be too small for the change in fCO2 at the sea surface at 18.25°C. It is smaller than the measurement precision for individual fCO2 measurements. - My overall impression is that the authors may well be underestimating the uncertainties, especially concerning the change in oceanic fCO2 between 1995-1997 and 2013-2015. Even if estimates of fCO2ocn for each of those 3-year periods can be made to within 3 $\mu$atm, the 2-sigma error bars for oceanic and atmospheric fCO2 would overlap. Furthermore, there has been no discussion of potential systematic errors nor their potential for evolution over time.

line 253: Such numbers should be given to at most one decimal point.

lines 290-291: - Delete "It is thus interesting to notice that". - Change "impact significantly" to "significantly affect".

line 296: I find that the error bar of +/-1.3 $\mu$mol/kg for the temporal change in DIC to be much too small. It is less than half of the measurement precision quoted by the authors.

These estimates are given to 4 significant figures when indeed it is not really justified to report them to better than 2 significant figures. The same holds for the numbers regported on line 298.

line 320: The uncertainty given for the annual average change in pH over the 18-year period is very small (0.0001) compared to estimates from other sites (aroung 0.0006). How do you explain this? Once again, it seems related to your use of SE instead of the standard deviation. The SE is misleading.

lines 337-338: Please provide support for this final sentence.

line 343: The authors need to bring up long-term (decadal) variability which is not addressed in this manuscript because sampling occured only over two 3-year windows and because a longer time series beyond 18 years may well be necessary.

line 348: The model study from Palmieri et al does not suggest a 15% contribution but rather a 25% contribution. Furthermore that model-based estimate is based on the anthropogenic carbon inventory in the Med Sea not on an estimated surface concentration of anthropogenic DIC. The relationship between the surface concentration and the vertical integral of the concentration (inventory) may not be one to one, and the difference between the two should be dstinguished in this study.

Global changes:

- Please make global changes so that there is always a space between all numbers and their units, e.g., 5 $\mu$atm, not 5$\mu$atm (line 98) and "3 m and 10 m" instead of "3m and 10m" (line 146).

- Please be consistent in your use of the abbreviation to represent total dissolved inorganic carbon. Sometimes you use DIC; other times you use TCO2. Actually, I would prefer to see the more modern abbreviation of CT, with T given as a subscript. For consistency, I would further recommend to use AT (with T also subscripted) for total alkalinity.

[Figure]

- Often citations in the text are provided with the wrong format. For example on lines 126-127 it says "using the equation of [Takahashi et al., 1993]". The square brakets are misplaced. If you are using the LaTeX template with BibTeX for Biogeosciences, this problem is easily fixed (use \citet instead of \citep).

REFERENCES

Byrne, R. H., Mecking, S., Feely, R. A., & Liu, X. (2010). Direct observations of basin‐wide acidification of the North Pacific Ocean. Geophysical Research Letters, 37(2).

Dickson, A. G., Sabine, C. L., and Christian, J. R.: Guide to best practices for ocean CO2 measurements, PICES Special Publication 3, 191 pp., 2007.

Landschützer, P., N. Gruber, F. Alexander Haumann, C. Rödenbeck, D. C. E. Bakker, S. van Heuven, Mario Hoppema, N. Metzl, C. Sweeney, T. Takahashi, B. Tilbrook, R. Wanninkhof (2015). The reinvigoration of the Southern Ocean carbon sink. Science, 349(6253), 1221-1224.

McKinley, G. A., Fay, A. R., Takahashi, T., & Metzl, N. (2011). Convergence of atmospheric and North Atlantic carbon dioxide trends on multidecadal timescales. Nature Geoscience, 4(9), 606.

---

## Author Comment (AC1) · 5 Dec 2017

Reviewer 2.def.docx- December, 5, 2017 Interactive comment on "Increase of dissolved inorganic carbon and decrease of pH in near surface waters of the Mediterranean Sea during the past two decades" by Liliane Merlivat et al. Anonymous Referee #2

In this manuscript, Merlivat et al. report on measurements of fCO2 during two 3-year windows whose midpoints are 18 years apart with samples taken adjacent locations in the Mediterranean Sea. They then combine those measurements with total alkalinity derived from measured temperature and salinity to compute DIC and pH. Because their derived DIC increase is larger than expected from equilibrium with atmospheric CO2, the authors invoke lateral transport of anthropogenic DIC from the Atlantic to the Mediterranean Sea to explain the difference.

GENERAL COMMENTS The authors report on quality measurements of fCO2, the fruit of decades of investement to develop and deploy the CARIOCA buoys with fCO2 sensors. They use the same measurement system for all measurements, thus allowing an assessement of the total change in ocean fCO2 between the 2 time periods that seems as precise as can be hoped for. Yet despite the quality of the measurements, my impression is that the uncertainties are underestimated when the authors discuss temporal changes in measured fCO2 as well as derived DIC and pH. This impression comes partly from the authors' choice to represent uncertainties as the standard error of the mean rather than the standard deviation. Their estimated uncertainties for the difference between these two time periods is usually much smaller than the best measurement precision. For more about my concerns on the uncertainty analysis of the authors, please see the detailed comments below, e.g., those labeled line 214, lines 243-248, line 296, and line 320. We bring details under these comments. An even greater concern is that the authors assume that the total temporal change is entirely anthropogenically driven. They do not consider the potential contribution from natural variability (see detailed comments below for the section commenting on 'lines 44-46:') The reviewer is right. A strong interannual variability of winter convection events between the two studied periods has been observed and must be taken into account to interpret the total temporal change of the computed increase of DIC. This is detailed in paragraph 4.3. . Because of these two concerns, it appears to me that the manuscript may well require in-depth revisions before it is acceptable for publication.

DETAILED COMMENTS lines 44-46: This statement from the authors in the introduction is an important one, making the point that there is large natural variability. Why then do they neglect to consider that natural decadal scale variabilty may explain part of the change between 1995-1997 and 2013-2017. In the North Atlantic, for instance it has been shown that because of decadal variability it requires 25 years for the long-term trend to emerge (McKinley et al, 2011). In the North Pacific, about half of the change in near surface ocean pH over a 15-year period has been ascribed to natural (non-anthropogenic) contributions (Byrne et al., 2010). In the Southern Ocean, early studies suggested a weaking of the Southern Ocean $CO_2$ uptake, but more recent work with 30-year perspective indicates a tendency in the opposite direction, with such oscillations being ascribed in part to natural variability (Lanschutzer, 2015). In contrast to these studies, the authors do not consider any contribution of natural decadal vari-ability in their interpretation, assigning the measured and estimated changes entirely to an anthrogenically forced trend. The change between the 2 points in time, even if they represent 3-year averages as in this study by Merlivat et al., are also likely to be affected by natural variability. This point is now discussed in paragraph 4.3 lines 53-55: - please add "over extended periods" after air-water interface – please delete "related to the absorption of increasing atmospheric $CO_2$ concentration" or nuance the mes-sage so as not to neglect natural variability. This has been done. lines 58-59: - please delete the commas just after "temperature" and just after "salinity" as these confuse the listing, making it appear longer that it is. You may also want add parentheses around 'T' and 'S', although I don't think that is necessary. This has been done. lines 76-77: - Can you provide references to support your statement that the Ligurian current isolates the two stations from coastal inputs. I would expect that eddies and jets would allow some transfer of heat, salt, momentum, and chemical species from coastal waters to the open Mediterranean Sea, even if that transfer is not occurring immediately adjacent to the two sampling sites. This has been well documented in Antoine et al, 2008, He-imburger et al, 2013 in addition to the work of Millot, 1999. - You could strengthen your case that the 2 stations (BOUSSOLE and DYFAMED) sample the same water mass by showing carbonate system measurements as well as T and S taken at the same time at both stations. line 83: change "They" to "Both" This has been done. lines 96-98: - add "K1 and K2" before "dissociation constants" - Why do the authors choose to use the K1 and K2 from Dickson and Millero (1987) even though the first author of the paper, when asked, suggests that there is a mistake in those formulations? I think it would be better to use K1 and K2 from Lueker et al. (2000), which is recommended for best practices (Dickson et al., 2007). We have kept the dissociation constants of Mehrbach refitted by Dickson and Millero [Dickson and Millero, 1987; Mehrbach et al., 1973] in order to remain consistent with the work previously published on Dyfamed [Begovic and Copin-Montegut, 2002; Copin-Montegut and Begovic, 2002] as one goal of our work was to compare data measured in close locations 18 years apart. However, we have checked that the computed DIC and pH changes deduced from a given change of fCO2 is identical when we consider one or the other set of constants. line 106: This sentence could be ambiguous. Are you referring to the standard deviation of the all 56 samples? Please clarify. This has been done. line 110: The authors use the term "fCO2@13" before it has been defined. Would it not be simpler just to delete "and fCO2@13" and get to the details later. This has been done. lines 120-121: The fCO2 is also a function of total dissolved inorganic phosphorus and silicon, when computed from DIC and total alkalinity, although in the oligotrophic surface waters of the Med Sea those nutrient concentrations are negligible and do not contribute significantly. We have modified the sentence. line 123: Did Takahashi et al. (1993) study the Med Sea? If not, how do you make the connection. The reference to Takahashi (1993) should not have been at that place. It has been deleted. line 130: change "decay of" to "decline in" This has been done. line 131: You could improve sentence flow by adding add "the ensuing" before "increase. This has been done. lines 134-135: The authors should provide evidence for their statement that the contribution of the air-sea flux is insignificant. This is well discussed for the years 1998-2000 in Begovic and Copin-Montegut ,2002 .For the period 2013-2015, the air –sea flux is equal to -0.45mmolm-2d-1, a value close to what was observed in previous years. This is indicated in the manuscript. line 140: change "15th to 26th" to "15 to 26". This has been done. line 142: The meaning of "Likewise" is not clear. Please modify sentence to clarify your meaning. This has been done . line 201 (211): The word "monotonous" means "boring" in English, perhaps not what was intended. I would suggest to use "monotonic" instead. OK. This has been done.
line 214: We have rewritten this part lines 237 to 244.We hope it is clearer now. - By "standard error" I presume that the authors are using the 'standard error of the mean', the latter 3 words which should be added to make it clearer to readers. We should have written Âń The mean value of dfCO2@13 is equal to 33.17 $\mu$atm with a standard error of the mean equal to 1.68$\mu$atm. Âż. In the original manuscript, we had computed the standard error of the mean equal to 6.29/$\sqrt{14}$=1.68 $\mu$atm , the standard deviation of the 14 values of dfCO2@13 being equal to 6.29 $\mu$atm. The standard deviation (SD) is a measure of variability. The standard error of the mean depends on both the standard deviation and the sample size. I have several problems with the authors' choice to use the standard error of the mean (SE) in this case. We agree with the reviewer that the error estimate in the previous version was confused as we did not separate accuracy and precision. In the new version, we consider the analytical accuracy of each sensor (2 $\mu$atm), as derived from the error on each sensor calibration and which has been confirmed experimentally by ship comparisons. This is now detailed section 2.2 and in lines 237 to 244. * First it gives the wrong impression that the uncertainty of these calculations is small (1.7 $\mu$atm), even lower than the precision of individual fCO2 measurements (3 $\mu$atm). Because the SE is the standard deviation divided by the square root of N, it is nearly 5 times smaller than the standard deviation in this case (N=24, Table 1). * Second, the result for the SE will also depend on the authors' arbitrary choice for the scale. * Third, even if the SE were appropriate, I do not understand how the authors get N=24 for the 'daily scale' mentioned in Table 1. This was a mistake. We intended to make subsampling but dividing N by 24 was not correct. It has been deleted. * Fourth, The use of SE in the right hand portion of Table 1 is at least visually inconsistent with the use of the standard deviation for each of the time periods shown in the left and center portions of the same table. I would stongly recommend that the authors simply use the standard deviation at least in Table 1. This has been done. If the authors insist on using SE, I would ask that they also provide the standard deviation in addition to the SE and that they statistically justify the use of the SE while explaining their choices in detail (e.g., N=24). There have been comments in scientific journals about the misuse of SE being a common practice. The SE could perhaps be used correctly here if well justified, but it can also mislead readers. line 215: The text says that "fCO2@13 is evenly distributed *in* the whole range of temperature". I am not sure I understand. It is seen in Table 1 that fCO2@13 varies from 19 to 45. Please clarify this sentence. We have modified the sentence and write "The distribution of values around the mean seems random and indicates no trend". line 217: Change "2 last decades" to "last two decades". This has been done. lines 228: You say that pH is on the seawater scale but later you use pHT, meaning it is on the total scale. Please clarify. We compute pH on the seawater scale. We delete T .We indicate in the text that the change of pH is computed at the mean in situ temperature 18.25°C line 231: The text says, "We used these sensitivity factors to compute the increase in DIC, ..." It is not clear why you need these sensitivity factors. Can you not simply compute DIC and pH for both time periods then take the difference? This has been changed. We just compute DIC and pH as suggested. line 232: The numbers for the increase in DIC are given with too many significant figures. We think it is coherent regarding the annual data reported for surface time series like for instance in [Bates et al., 2014]. Table 2: The numbers for dfCO2 and dDIC are given with 4 significant figures, much too much. The number of significant figures given in the paper is often too many. The authors should carefully go over the reported numbers and reduce to a justifiable number of significant figures in every case. We keep two significant figures for the annual change data being coherent with numbers reported for surface time series like for instance in [Bates et al., 2014] . lines 243-248: - Please inform the reader what the error bars are reporting, standard deviation or standard error of the mean. There is insufficient information about how 'atmospheric fCO2' was calculated from atmospheric xCO2. Did the authors make a humidity correction, which can change numbers by a few percent? Nothing along those lines was mentioned. How much of a difference would there be if the authors did not assume that the atmospheric pressure is 1 atm. Did they make the xCO2-to fCO2 conversion on a monthly basis and then take an annual average? Currently it seems they are making only an annual-mean calculation. Would results differ? – fCO2 atmwas computed as:

with x CO2 molar fraction of CO2 in the atmosphere , pH2O at 18.25°C equal to 21mb , P equal to 1013mb, and f, factor to convert partial pressure to fugacity, equal to 0.9966. Then: fCO2=0.976 xCO2. For a sensitivity test, as a meteorological buoy was in place close to the mooring during the 2013-2015 period, we have made the same exercice taking into account the monthly distribution of x, pH2O and P. We get the same factor to convert xCO2 in fCO2 as when considering annual values. The mean annual value of fCO2 $\mu$atm is computed as follows considering monthly values of xCO2: 1995-1997: fCO2mean=355.3 $\mu$atm, N=36 , SD=5.0, SE=0.8. 2013-2015: fCO2mean=389.6 $\mu$atm, N=36 , SD=5.5, SE=0.9. We then calculate: dfCO2 =34.3+/-2.3 $\mu$atm with SE=1.2.

The error estimate appears to be too small for the change in fCO2 at the sea surface at 18.25 C. It is smaller than the measurement precision for individual fCO2 measurements. - My overall impression is that the authors may well be underestimating the uncertainties, especially concerning the change in oceanic fCO2 between 1995-1997 and 2013-2015. Even if estimates of fCO2ocn for each of those 3-year periods can be made to within 3 $\mu$atm, the 2-sigma error bars for oceanic and atmospheric fCO2 would overlap. Furthermore, there has been no discussion of potential systematic errors nor their potential for evolution over time. line 253 : Such numbers should be given to at most one decimal point. We have made changes. lines 290-291: - Delete "It is thus interesting to notice that". - Change "impact significantly" to "significantly affect". This has been done. line 296: I find that the error bar of +/-1.3 $\mu$mol/kg for the temporal change in DIC to be much too small. It is less than half of the measurement precision quoted by the authors. These estimates are given to 4 significant figures when indeed it is not really justified to report them to better than 2 significant figures. The same holds for the numbers reported on line 298. Changes have been made. line 320: The uncertainty given for the annual average change in pH over the 18-year period is very small (0.0001) compared to estimates from other sites (aroung 0.0006). How do you explain this? Once again, it seems related to your use of SE instead of the standard deviation. The SE is misleading. Our number (0.0001) is very comparable to other data reported in the literature. For instance, Bates et al (2014) in the analysis of 7 pH time series indicate standard error changes of pH of 0.0001 for the BATS and HOT sites and 0.0002 for ESTOC lines 337-338: Please provide support for this final sentence. We have added the value of the Revelle factor close to 10 and deleted the last sentence. line 343: The authors need to bring up long-term (decadal) variability which is not addressed in this manuscript because sampling occured only over two 3-year windows and because a longer time series beyond 18 years may well be necessary. We have modified the sentence. line 348: The model study from Palmieri et al does not suggest a 15% contribution but rather a 25% contribution. OK Furthermore that model-based estimate is based on the anthropogenic carbon inventory in the Med Sea not on an estimated surface concentration of anthropogenic DIC. The relationship between the surface concentration and the vertical integral of the concentration (inventory) may not be one to one, and the difference between the two should be dstinguished in this study. It is exact that vertical profiles of anthropogenic carbon in the Med Sea indicate higher concentration of anthropogenic carbon in the upper part or the water column (Huertas et al,2009, Schneider et al, 2010). However both studies establish that there is a net flux of anthropogenic carbon from the Atlantic towards the Mediterranean basin. (Schneider et al, 2010) propose that it may represent about 10% of the total inventory of Cant in the whole basin. We have corrected the sentence in our text.

Global changes: - Please make global changes so that there is always a space between all numbers and their units, e.g., 5 $\mu$atm, not 5$\mu$atm (line 98) and "3 m and 10 m" instead of "3m and 10m" (line 146). Corrections have been made.. - Please be consistent in your use of the abbreviation to represent total dissolved inorganic carbon. Sometimes you use DIC; other times you use TCO2. Actually, I would prefer to see the more modern abbreviation of CT, with T given as a subscript. For consistency, I would further recommend to use AT (with T also subscripted) for total alkalinity. We have deleted TCO2.We use DIC and Alk - Often citations in the text are provided with the wrong format. For example on lines 126-127 it says "using the equation of [Takahashi et al., 1993]". The square brakets are misplaced. If you are using the LaTeX template with BibTeX for Biogeosciences, this problem is easily fixed (use \citet instead of \citep). We will check carefully in the manuscript. REFERENCES Byrne, R. H., Mecking, S., Feely, R. A., & Liu, X. (2010). Direct observations of basinâ A RËǦ wide acidification of the North Pacific Ocean. Geophysical Research Letters, 37(2). Dickson, A. G., Sabine, C. L., and Christian, J. R.: Guide to best practices for ocean CO2 measurements, PICES Special Publication 3, 191 pp., 2007. Landschützer, P., N. Gruber, F. Alexander Haumann, C. Rödenbeck, D. C. E. Bakker, S. van Heuven, Mario Hoppema, N. Metzl, C. Sweeney, T. Takahashi, B. Tilbrook, R. Wanninkhof (2015). The reinvigoration of the Southern Ocean carbon sink. Science, 349(6253), 1221-1224. McKinley, G. A., Fay, A. R., Takahashi, T., & Metzl, N. (2011). Convergence of atmospheric and North Atlantic carbon dioxide trends on multidecadal timescales. Nature Geoscience, 4(9), 606. Interactive comment on Biogeosciences Discuss., https://doi.org/10.5194/bg-2017-284, 2017.

Bates, N., Y. Astor, M. Church, K. Currie, J. Dore, M. Gonaález-Dávila, L. Lorenzoni, F. Muller-Karger, J. Olafsson, and M. Santa-Casiano (2014), A Time-Series View of Changing Ocean Chemistry Due to Ocean Uptake of Anthropogenic CO2 and Ocean Acidification, Oceanography, 27(1), 126-141. Begovic , M., and C. Copin-Montegut (2002), Processes controlling annual variations in the partial pressure of fCO2 in surface waters of the central northwestern Mediterranean sea (Dyfamed site), Deep-Sea Research II, 49, 2031-2047. Copin-Montegut, C., and M. Begovic (2002), Distributions of carbonate properties and oxygen along the water column (0–2000 m) in the central part of the NW Mediterranean Sea (Dyfamed site): influence of winter vertical mixing on air–sea CO2 and O2 exchanges, Deep-Sea Research II 49, 2049-2066. Dickson, A. G., and F. J. Millero (1987), A comparison of the equilibrium constants for the dissociation of carbonic acid in seawater media, Deep Sea Research Part A. Oceanographic Research Papers, 34(10), 1733-1743. Mehrbach, C., C. H. Culberson, J. E. Hawley, and R. M. Pytkowicx (1973), Measurement of the apparent dissociation constants of carbonic acid in seawater at atmospheric pressure, Limnol

Oceanogr, 18(6), 897-907.

Please also note the supplement to this comment:
https://www.biogeosciences-discuss.net/bg-2017-284/bg-2017-284-AC1-
supplement.pdf

**Supplement:**

[revised manuscript text omitted]

---

## Author Comment (AC2) · 5 Dec 2017

Please note the corrected version of the reply to the comment : lines 134-135: The authors should provide evidence for their statement that the contribution of the air-sea flux is insignificant. This is well discussed for the years 1998-2000 in Begovic and Copin-Montegut, 2002 .For the period 2013-2015, the air –sea flux is equal to -0.45mol m-2yr-1, a value close to what was observed in previous years. This is indicated in the manuscript. In the previous document it was written -0.45 mmol m-2 d-1

---

## Author Comment (AC3) · 7 Dec 2017

Reviewer 1 1 .def.docx-December, 7, 2017. 2 Interactive comment on "Increase of dissolved inorganic carbon and decrease of pH in near 3 surface waters of the Mediterranean Sea during the past two decades" by Liliane Merlivat et 4 al. 5 6 Anonymous Referee #1 7 8 9 The paper by Merlivat et al. provides a description of carbonate chemistry in two close fixed 10 station located in the Ligurian Sea (northwestern Mediterranean Sea). By combining time 11 series data of CO2 fugacity with alkalinity derived estimations, they reported an increase of 12 dissolved inorganic carbon and decrease of pH in near surface waters during the past two 13 decades. This issue is of particular interest to the referee and I think that the authors have a 14 very nice data set to exploit. However, I think the analysis is somewhat incomplete, and I 15 finished the paper wanted a more in-depth analysis and discussion. I encourage the authors to 16 further expand their work because at this stage their hypothesis are not well supported. The 17 manuscript could be published in Biogeosciences after a major revision in order to clarify 18 some aspects as indicated below. 19 20 Major Comments 21 My major reservation about this work is the difference between the measured fCO2 at the sea 22 surface (fCO2sea) and the fCO2 derived from atmospheric xCO2 concentration (fCO2air). In 23 2013-2015 the sea surface mean annual fCO2 calculated at 18.25_C (the mean annual in situ 24 temperature) was larger than the fCO2air derived from atmospheric data at the same 25 temperature. This result is quite strange, because it means a CO2 outgassing from the sea 26 surface to the atmosphere on annual average, which is in contrast with respect to the ongoing 27 ocean acidification process and the general net anthropogenic CO2 uptake measured in the 28 Mediterranean Sea by different research. 29 In 2013-2015 I would expect an equilibrium between the fCO2sea and fCO2air, or a slightly 30 higher value in the fCO2air, as it was detected in the 1995-1997. How the authors can explain 31 this issue? 32 In the 2 periods, 1995-1997 and 2013-2015, the CO2 annual flux is directed from the 33 atmosphere to the sea in both cases, although the annual average of fCO2 in surface seawater 34 in 2013-2015 is higher than atmospheric fCO2. This is due to higher wind speed in autumn 2 and winter when the surface water is undersaturated. This is well illustrated 35 in the figure in 36 the attached file (Figure flux.pdf) for the time period 2013-2015. In the upper figure, the three 37 thin lines indicate fCO2 atm. 38 The mean annual CO2 flux is equal to -0.45 mol.m-2.yr-1 using the exchange coefficient of 39 [Wanninkhof, 2014]. 40 They suggested the contribution of the Atlantic Ocean as a source of anthropogenic carbon, 41 but I do not understand how the Atlantic surface waters can be relatively enriched in 42 anthropogenic carbon. 43 [Huertas et al., 2009] conducted a sampling program at eight fixed stations in the Strait of 44 Gibraltar to study natural and anthropogenic carbon exchange between the Atlantic Ocean and 45 the Mediterranean Sea. Their results show that Atlantic water has a higher concentration of 46 anthropogenic carbon than Mediterranean water. A decreasing vertical gradient of Cant in the 47 water column is observed, the upper layers being enriched in Cant (Figures 5 and 6). 48 Moreover, this is in contrast with the end of the discussion where the authors saythat 49 (P13L331) "The Mediterranean Sea is actually able to absorb more anthropogenic CO2 per 50 unit area". 51 As stated in the text, surface waters of the Mediterranean basin have a relatively low Revelle 52 factor, close to 10, due to a high alkalinity and a high temperature and therefore have a 53 relatively high uptake capacity for Cant. 54 Maybe there are other causes which could explain the fCO2 increase at the sea surface 55 observed in 2013-2015, such as a stronger and deeper winter vertical mixing with CO2 56 enriched LIW. 57 The reviewer is right. A strong interannual variability of winter convection events between 58 the two studied periods has been observed and must be taken into account to interpret the total 59 temporal change of the computed increase of DIC. This is detailed in paragraph 4.3, lines 323 60 -329. 61 Finally, additional information about the water mass exchange throughout the Strait of 62 Gibraltar and its temporal variation are needed. 63 This is analyzed and discussed in [Huertas et al., 2009], see for instance figure 7. See also 64 [Schneider et al., 2010], table 2. 65 These can be found in the recent review of Jordà et al. (2017) which may provide more 66 insights for this work. 67 The authors found a DIC increase larger than expected from equilibrium with atmospheric 3 CO2. They hypnotized a _15% contribution of the Atlantic 68 Ocean as a source of 69 anthropogenic carbon to the Mediterranean Sea through the strait of Gibraltar. I think that the 70 analysis presented in the manuscript are not sufficient to support such hypothesis and the 71 authors should provide a lot more analysis and discussions. 72 This is detailed in the paragraph 4.3. 73 Moreover, the Mediterranean Sea overturning circulation and the sites of dense water 74 formation could play a very important role in the sequestration of anthropogenic CO2 and in 75 the ocean acidification of the Mediterranean Sea.I think that the authors should read the recent 76 papers of Touratier et al. (2016), Ingrosso et al. (2017), and Krasakopoulou et al. (2017), who 77 estimated the anthropogenic CO2 in the Gulf of Lion, Adriatic Sea, and the Aegean Sea 78 respectively. 79 Certainly the reasons why the Mediterranean Sea water column stores large amounts of 80 anthropogenic CO2 are due to the fast deep water formation processes combined with surface 81 water having high potential to take up Cant due to a relatively low Revelle factor. 82 The authors try to assess the influence of physical and biological process on the seasonal and 83 inter-annual variation of fCO2. To do this, they used a simple analysis of the change of 84 fCO2@13 (fCO2 normalized to the constant temperature of 13_C) as a function of SST, 85 which is not sufficient to achieve the scope. I suggest to quantify (1) the air-sea CO2 86 exchange and (2) the thermal/not-thermal contributions on the fCO2 variation with the 87 method of Takahashi et al. (2002). In this way the authors could clarify how fCO2 seasonal 88 variation is affected by physical (i.e. temperature, mixing, and air-sea CO2 exchange) and 89 biological processes (i.e. photosynthesis, respiration, and calcification). 90 The objective of our paper is to compare the time change of surface fCO2 measurements 91 made at 2 very close locations, Dyfamed and Boussole, at an interval of 18 years. The 92 processes that govern the distribution of fCO2 at the annual scale at the same site have been 93 analyzed in detail in a publication entitled "Processes controlling annual variations in the 94 partial pressure of CO 2 in surface waters of the central northwestern Mediterranean Sea 95 (Dyfamed site)[Begovic and Copin-Montegut, 2002]. For instance, the figure 8 in this paper 96 is a good illustration of the relative importance of individual processes which govern the 97 distribution of DIC over an annual cycle. For this reason, we decided not to repeat this well98 argued description which is already published. 99 100 Specific Comments 101 P4L93: If the authors followed the standard operational procedures, the reference of Dickson 4 102 et al. (2007) could be added to Edmond (1970). 103 The reference to Edmond (1970) is line 102. 104 P5L126: I propose to consider here the the method of Takahashi et al. (2002) and to present 105 the temporal variation of the thermal and not-thermal fCO2 as differences (dfCO2) with 106 respect to the February, chosen
as reference month because it usually presents the lowest 107 temperature and the minimum biological activity. 108 We have chosen to estimate the difference between the values of the thermal component 109 fCO2@13 two decades apart according to the temperature (14 temperature steps of 1°) and 110 not to the time. This approach is more quantitative than a comparison of monthly values 111 because we know that key processes which control the fCO2@13 distribution such as the 112 beginning of the bloom depend more directly on a narrow temperature threshold (13-14 °) 113 while it may vary up to one month. 114 P5L128: The "remineralization" is a biological activity. Please modify/clarify the sentence. 115 This has been done (line 139). 116 P5L130: Do the authors have oxygen data? The examination of the O2/DIC or AOU 117 (apparent oxygen utilization)/DIC ratio would provide useful information about the influence 118 of biological activity to the observed fCO2 variation. Also satellite data of Chloro-Phyll phyll 119 a concentration may help, which nowadays are easy to get 120 See our comment above (lines 90-98 in this text). 121 P6L134: "The contribution of air-sea exchange is not significant". In order to support this 122 sentence, please can the authors calculate the air-sea CO2 flux and estimate the real influence 123 of this process? 124 This has been done, lines 146-148. 125 P6L150: Levantine Intermediate Water (LIW) originates in the Eastern Mediterranean and 126 takes years to reach the Ligurian Sea. Due to the organic matter remineralization processes, 127 the LIW presents low dissolved oxygen concentration and high CO2 levels (Álvarez et al., 128 2014), even higher than then the atmospheric levels. Taking into account these considerations, 129 in the present study, the increase of total dissolved inorganic carbon observed in 2013-2015 130 can be related to a stronger and deeper winter vertical mixing with CO2 enriched LIW? 131 The reviewer is right. A strong interannual variability of winter convection events between 132 the two studied periods has been observed and must be taken into account to interpret the total 133 temporal change of the computed increase of DIC. This is detailed in paragraph 4.3, lines 323 134 -329. 135 As reported by Alvarez et al. (2014), the LIW during its westward flows can increase DIC and 5 lower pHT 136 of different Mediterranean basin. 137 P7L197:

[Figure]

"mixing with enriched deep waters" please substitute with "mixing with CO2- 138 enriched deep waters". This may support the hypothesis of a general DIC increase generated 139 by mixing with LIW, but further analysis and more discussions are needed. 140 P8L199: During summer, due to the high sea surface temperature, the CO2 flux from the sea 141 to the atmosphere could also play an important role. Please consider also this process in 142 addition to the biological drawdown of carbon. 143 See our comment above (lines 90-98 in this text). 144 P9L223: "Changes of seawater carbonate chemistry in surface waters". This section needs 145 some modification/clarification. L223-227 seems more appropriate for the Material and 146 methods. 147 In Material and methods, we consider the DIC and Alk analysis of the seawater samples 148 taken at Boussole during the servicing cruises to the mooring. In the section 3.4, we consider 149 the derived values of DIC and pH from the analysis of the 2 time series of fCO2. 150 L229-234: DIC and pH are derived parameters. They are calculated from total alkalinity and 151 fCO2. Due to this reason, the fCO2-DIC and fCO2-pH may not have sense and the near 152 perfect R2 is not significant. Please, can the authors clarify this issue? 153 This has been changed. We just compute DIC and pH as suggested. 154 P9L229: pHT refers to the pH on the total scale. But the authors calculated the pH on the 155 seawater scale (P9L228) which is conventionally denoted as pHsws. Please substitute in all 156 the manuscript/figures the pHT with pHsws. 157 We compute pH on the seawater scale. We delete T .We indicate in the text that the change of 158 pH is computed at the mean in situ temperature 18.25°C 159 P11L259: Any references which can support that Atlantic surfacewaters are relatively 160 enriched in anthropogenic carbon and why? 161 See [Huertas et al., 2009]. 162 Even if the Atlantic surface water could be enriched in CO2, I do not think that it could 163 preserve this property. An air-sea equilibrium, mixing, and biological processes may happen 164 during the long time that Atlantic surface water spent to reach the Ligurian Sea from the 165 Gibraltar Strait. 166 The depth of the surface water layer of the Atlantic entering the Mediterranean Sea through 167 the Strait of Gibraltar is close to 200 meters. It would take a few months to reach the

Dyfamed zone assuming a lower estimate of the average current close to 10 cm / s on its route 169 along the Algerian coast and then northwards [Millot, 1999]. This indicates that CO2-enriched 6 Atlantic water may retain its signature during this 170 relatively short period of time. 171 P11L270-272: More discussion and references are needed to support this sentence. 172 This was not correct. As indicated earlier, and illustrated in the figure, although the annual 173 average of fCO2 in surface sea water was higher than atmospheric fCO2, the annual flux was 174 directed from the atmosphere to the sea. 175 P13L335: More appropriate and recent references are Touratier et al. (2016), Ingrosso et al. 176 (2017), and Krasakopoulou et al. (2017), who estimated the anthropogenic CO2 in the three 177 dense water formation area of the Mediterranean Sea. 178 We believe that the 2 references cited [Schneider et al., 2010]and [Palmiéri et al., 2015] give 179 the relevant information in relation to the western basin of the Mediterranean Sea which is 180 studied in our paper. 181 182 Technical comments 183 I suggest to improve the general quality of the figures. 184 P11L286: "P=0,0749" Substitute the coma with point. This has been done. 185 References 186 Álvarez, M., Sanleón-Bartolomé, H., Tanhua, T., Mintrop, L., Luchetta, A., Cantoni, C., 187 Schroeder, K., Civitarese, G., 2014. The CO2 system in the Mediterranean Sea: a basin wide 188 perspective. Ocean Sci. 10, 69–92. 189 Carter, B. R., N. L. Williams, A. R. Gray, and R. A. Feely. 2016. Locally interpolated 190 alkalinity regression for global alkalinity estimation. Limnol. Oceanogr. Methods 14: 268– 191 277. doi:10.1002/lom3.10087 192 Dickson, A.G., Sabine, C.L., Christian, J.R., 2007. Guide to best practices for ocean CO2 193 measurements. PICES Spec. Publ. 3, 191. 194 Ingrosso, G., Bensi, M., Cardin, V., Giani, M., 2017. Anthropogenic CO2 in a dense water 195 formation area of the Mediterranean Sea. Deep-Sea Research Part I 123, 118– 128. 196 doi:10.1016/j.dsr.2017.04.004 197 Jordà, G., Schuckmann, Von, K., Josey, S.A., Caniaux, G., García-Lafuente, J., Sammartino, 198 S., Ozsoy, E., Polcher, J., Notarstefano, G., Poulain, P.M., Adloff, F., Salat, J., Naranjo, C., 199 Schroeder, K., Chiggiato, J., Sannino, G., Macías, D., 2017. The Mediterranean Sea heat and 200 mass budgets: Estimates, uncertainties and perspectives.

Progress in Oceanography 156, 201 174–208. doi:10.1016/j.pocean.2017.07.001 202 Krasakopoulou, E., Souvermezoglou, E., Giannoudi, L., Goyet, C., 2017. Carbonate system 203 parameters and anthropogenic CO2 in the North Aegean Sea during October 2013. 7 Continental 204 Shelf Research 1–13. doi:10.1016/j.csr.2017.04.002 205 Touratier, F., Goyet, C., Houpert, L., de Madron, X.D., Lefèvre, D., Stabholz, M., Guglielmi, 206 V., 2016. Deep-Sea Research I. Deep-Sea Research Part I 113, 33–48. 207 doi:10.1016/j.dsr.2016.04.003 208 Interactive comment on Biogeosciences Discuss., https://doi.org/10.5194/bg-2017-284, 2017. 209 210 References 211 Begovic , M., and C. Copin-Â∎‐Montegut (2002), Processes controlling annual variations in 212 the partial pressure of fCO2 in surface waters of the central northwestern 213 Mediterranean sea (Dyfamed site), Deep-Â∎Sea Research II, 49, 2031-Â∎‐2047. 214 Huertas, I. E., A. F. Ríos, J. García-Â∎‐Lafuente, A. Makaoui, S. ' Rodríguez-Â∎‐Gálvez, A. Sánchez-Â∎‐ 215 Román, A. Orbi, J. Ruíz, and F. F. and Pérez (2009), Anthropogenic and natural CO2 216 exchange through the Strait of Gibraltar, Biogeosciences, 6, 647-Â∎‐662. 217 Millot (1999), Circulation in the Western Mediterranean Sea, Journal of Marine Systems, 218 20, 423–442. 219 Palmiéri, J., J. C. Orr, J. C. Dutay, K. Béranger, A. Schneider, J. Beuvier, and S. Somot 220 (2015), Simulated anthropogenic CO2 storage and acidification of the Mediterranean 221 Sea, Biogeosciences, 12(3), 781-Â∎‐802. 222 Schneider, A., T. Tanhua, A. Körtzinger, and D. W. R. Wallace (2010), High anthropogenic 223 carbon content in the eastern Mediterranean, Journal of Geophysical Research, 115(C12). 224 Wanninkhof, R. (2014), Relationship between wind speed and gas exchange over the 225 ocean revisited, Limnology and Oceanography: Methods, 12(6), 351-Â∎‐362. 226 227

Please also note the supplement to this comment:
https://www.biogeosciences-discuss.net/bg-2017-284/bg-2017-284-AC3-supplement.pdf
* * *
[Figure]

**Fig. 1.**

**Supplement:**

[revised manuscript text omitted]

---

## Referee Report (RR1)

**Major Comments**

My major reservation about this work is the difference between the measured fCO2 at the sea surface (fCO2sea) and the fCO2 derived from atmospheric xCO2 concentration (fCO2air). In 2013-2015 the sea surface mean annual fCO2 calculated at 18.25_C (the mean annual in situ temperature) was larger than the fCO2air derived from atmospheric data at the same temperature. This result is quite strange, because it means a CO2 outgassing from the sea surface to the atmosphere on annual average, which is in contrast with respect to the ongoing ocean acidification process and the general net anthropogenic CO2 uptake measured in the Mediterranean Sea by different research. In 2013-2015 I would expect an equilibrium between the fCO2sea and fCO2air, or a slightly higher value in the fCO2air, as it was detected in the 1995-1997. How the authors can explain this issue?

In the 2 periods, 1995-1997 and 2013-2015, the $CO_2$ annual flux is directed from the atmosphere to the sea in both cases, although the annual average of $fCO_2$ in surface seawater in 2013-2015 is higher than atmospheric $fCO_2$. This is due to higher wind speed in autumn and winter when the surface water is undersaturated. This is well illustrated in the figure below for the time period 2013-2015. In the upper figure, the three thin lines indicate $fCO_2$ atm.

This could be a good explanation, but it must be supported by a statistical analysis of the data. Is there a significant statistical difference in the wind speed between winter/spring/summer/autumn? From the the figure proposed, the wind speed seems more or less the same during the different month.

The mean annual $CO_2$ flux is equal to -0.45 mol $.m^{-2}.yr^{-1}$ using the exchange coefficient of [*Wanninkhof*, 2014].

How was calculated the mean annual CO2 flux? If this is the average of the

daily CO2 flux, it is also necessary to report the standard deviation. Please clarify.

They suggested the contribution of the Atlantic Ocean as a source of anthropogenic carbon, but I do not understand how the Atlantic surface waters can be relatively enriched in anthropogenic carbon.

[*Huertas et al.*, 2009] conducted a sampling program at eight fixed stations in the Strait of Gibraltar to study natural and anthropogenic carbon exchange between the Atlantic Ocean and the Mediterranean Sea. Their results show that Atlantic water has a higher concentration of anthropogenic carbon than Mediterranean water. A decreasing vertical gradient of Cant in the water column is observed, the upper layers being enriched in Cant (Figures 5 and 6).

My doubts remain. Since Cant cannot be measured directly, as it cannot be chemically discriminated from the bulk of dissolved inorganic carbon, different approaches for its indirect estimation have been developed. All the proposed approach do not give good results in the surface layer, due to the effect of the biological activity and the strong dynamic of this portion of the water column. For these reasons usually the the surface waters (0-200m) is not considered in the estimation of Cant. Touratier et al. (2012) strongly criticized *Huertas et al.*, 2009 to calculate the Cant in the surface layer, and Palmieri et al. (2015) also reported Cant calculation of the surface layer. So, is the Atlantic Ocean a sink or a source of Can respect to the Mediterranean Sea? At the moment we do not have clear scientific evidence to answer at this question.

Moreover, this is in contrast with the end of the discussion where the authors saythat (P13L331) "The Mediterranean Sea is actually able to absorb more anthropogenic CO2 per unit area".

As stated in the text, surface waters of the Mediterranean basin have a relatively low Revelle factor, close to 10, due to a high alkalinity and a high temperature and therefore have a relatively high uptake capacity for Cant.

The answer is not pertinent to may question. I try to be more clear. How the Atlantic Ocean can be a source of the Can if (as the authors say P13L331)

"The Mediterranean Sea is actually able to absorb more anthropogenic CO2 per unit area"?

Maybe there are other causes which could explain the fCO2 increase at the sea surface observed in 2013-2015, such as a stronger and deeper winter vertical mixing with CO2 enriched LIW.

The reviewer is right. A strong interannual variability of winter convection events between the two studied periods has been observed and must be taken into account to interpret the total temporal change of the computed increase of DIC. This is detailed in paragraph 4.3, lines 323 -329.

Finally, additional information about the water mass exchange throughout the Strait of Gibraltar and its temporal variation are needed.

This is analyzed and discussed in [*Huertas et al*., 2009], see for instance figure 7. See also [*Schneider et al*., 2010], table 2.

These can be found in the recent review of Jordà et al. (2017) which may provide more insights for this work.  The authors found a DIC increase larger than expected from equilibrium with atmospheric CO2. They hypnotized a _15% contribution of the Atlantic Ocean as a source of anthropogenic carbon to the Mediterranean Sea through the strait of Gibraltar. I think that the analysis presented in the manuscript are not sufficient to support such hypothesis and the authors should provide a lot more analysis and discussions.

This is detailed in the paragraph 4.3.

Why the author do not consider the recent review of Jordà et al. (2017) about the water mass exchange in the Strait of Gibraltar?

Moreover, the Mediterranean Sea overturning circulation and the sites of dense water formation could play a very important role in the sequestration of anthropogenic CO2 and in the ocean acidification of the Mediterranean Sea.I think that the authors should read the recent papers of Touratier et al. (2016), Ingrosso et al. (2017), and Krasakopoulou et al. (2017), who estimated the anthropogenic CO2 in the Gulf of Lion, Adriatic Sea, and the Aegean Sea

respectively.

Certainly the reasons why the Mediterranean Sea water column stores large amounts of anthropogenic $CO_2$ are due to the fast deep water formation processes combined with surface water having high potential to take up Cant due to a relatively low Revelle factor.

Ok, but why the author do not want to consider and to cite these recent papers which estimate the Cant in Mediterranean Sea? Touratier et al. (2016) also estimate the Cant in an area very close respect to the DYFAMED site.

The authors try to assess the influence of physical and biological process on the seasonal and inter-annual variation of fCO2. To do this, they used a simple analysis of the change of fCO2@13 (fCO2 normalized to the constant temperature of 13_C) as a function of SST, which is not sufficient to achieve the scope. I suggest to quantify (1) the air-sea CO2 exchange and (2) the thermal/not-thermal contributions on the fCO2 variation with the method of Takahashi et al. (2002). In this way the authors could clarify how fCO2 seasonal variation is affected by physical (i.e. temperature, mixing, and air-sea CO2 exchange) and biological processes (i.e. photosynthesis, respiration, and calcification).

The objective of our paper is to compare the time change of surface $fCO_2$ measurements made at 2 very close locations, Dyfamed and Boussole, at an interval of 18 years. The processes that govern the distribution of $fCO_2$ at the annual scale at the same site have been analyzed in detail in a publication entitled "Processes controlling annual variations in the partial pressure of CO$_2$ in surface waters of the central northwestern Mediterranean Sea (Dyfamed site)[*Begovic and Copin-Montegut*, 2002]. For instance, the figure 8 in this paper is a good illustration of the relative importance of individual processes which govern the distribution of DIC over an annual cycle. For this reason, we decided not to repeat this well- argued description which is already published.

**Specific Comments**

P4L93: If the authors followed the standard operational procedures, the

reference of Dickson et al. (2007) could be added to Edmond (1970).

The reference to Edmond (1970) is line 102.

Where is the reference of Dickson et al. (2007)? Did the authors follow the standard operational procedures?

P5L126: I propose to consider here the the method of Takahashi et al. (2002) and to present the temporal variation of the thermal and not-thermal fCO2 as differences (dfCO2) with respect to the February, chosen as reference month because it usually presents the lowest temperature and the minimum biological activity.

We have chosen to estimate the difference between the values of the thermal component fCO2@13 two decades apart according to the temperature (14 temperature steps of 1°) and not to the time. This approach is more quantitative than a comparison of monthly values because we know that key processes which control the fCO2@13 distribution such as the beginning of the bloom depend more directly on a narrow temperature threshold (13-14 °) while it may vary up to one month.

P5L128: The "remineralization" is a biological activity. Please modify/clarify the sentence.

This has been done (line 139).

P5L130: Do the authors have oxygen data? The examination of the O2/DIC or AOU (apparent oxygen utilization)/DIC ratio would provide useful information about the influence of biological activity to the observed fCO2 variation. Also satellite data of Chloro-Phyll phyll a concentration may help, which nowadays are easy to get

See our comment above before Specific Comments.

Do the authors have oxygen data? I do not found answer to this question.

P6L134: "The contribution of air-sea exchange is not significant". In order to support this sentence, please can the authors calculate the air-sea CO2 flux and estimate the real influence of this process?

This has been done, lines 146-148.

P6L150: Levantine Intermediate Water (LIW) originates in the Eastern Mediterranean and takes years to reach the Ligurian Sea. Due to the organic matter remineralization processes, the LIW presents low dissolved oxygen concentration and high CO2 levels (Álvarez et al., 2014), even higher than then the atmospheric levels. Taking into account these considerations, in the present study, the increase of total dissolved inorganic carbon observed in 2013-2015 can be related to a stronger and deeper winter vertical mixing with CO2 enriched LIW?

The reviewer is right. A strong interannual variability of winter convection events between the two studied periods has been observed and must be taken into account to interpret the total temporal change of the computed increase of DIC. This is detailed in paragraph 4.3, lines 323 -329.

As reported by Alvarez et al. (2014), the LIW during its westward flows can increase DIC and lower pHT of different Mediterranean basin. P7L197: "mixing with enriched deep waters" please substitute with "mixing with CO2- enriched deep waters". This may support the hypothesis of a general DIC increase generated by mixing with LIW, but further analysis and more discussions are needed.

No reply to this comment.

P8L199: During summer, due to the high sea surface temperature, the CO2 flux from the sea to the atmosphere could also play an important role. Please consider also this process in addition to the biological drawdown of carbon.

See our comment above before Specific Comments

I do not understand why the author do not consider the influence of the CO2 flux from the sea to the atmosphere.

P9L223: "Changes of seawater carbonate chemistry in surface waters". This section needs some modification/clarification. L223-227 seems more appropriate for the Material and methods.

In Material and methods, we consider the DIC and Alk analysis of the

seawater samples taken at Boussole during the servicing cruises to the mooring. In the section 3.4, we consider the derived values of DIC and pH from the analysis of the 2 time series of fCO$_2$.

L229-234: DIC and pH are derived parameters. They are calculated from total alkalinity and fCO2. Due to this reason, the fCO2-DIC and fCO2-pH may not have sense and the near perfect R2 is not significant. Please, can the authors clarify this issue?

This has been changed. We just compute DIC and pH as suggested.

P9L229: pHT refers to the pH on the total scale. But the authors calculated the pH on the seawater scale (P9L228) which is conventionally denoted as pHsws. Please substitute in all the manuscript/figures the pHT with pHsws.

We compute pH on the seawater scale. We delete T .We indicate in the text that the change of pH is computed at the mean in situ temperature 18.25°C

You should substitute in all the manuscript/figures the pHT with pHsws. Not only delete T. Only pH is not correct and ambiguous.

P11L259: Any references which can support that Atlantic surface waters are relatively enriched in anthropogenic carbon and why?

See [*Huertas et al*., 2009].

In the same paper the TrOCA approach measured a greater Cant in the Mediterranean waters.

Even if the Atlantic surface water could be enriched in CO2, I do not think that it could preserve this property. An air-sea equilibrium, mixing, and biological processes may happen during the long time that Atlantic surface water spent to reach the Ligurian Sea from the Gibraltar Strait.

The depth of the surface water layer of the Atlantic entering the Mediterranean Sea through the Strait of Gibraltar is close to 200 meters. It would take a few months to reach the Dyfamed zone assuming a lower estimate of the average current close to 10 cm / s on its route along the Algerian coast and then northwards [*Millot*, 1999]. This indicates that CO$_2$-

enriched Atlantic water may retain its signature during this relatively short period of time.

P11L270-272: More discussion and references are needed to support this sentence.

This was not correct. As indicated earlier, and illustrated in the figure, although the annual average of $fCO_2$ in surface seawater was higher than atmospheric $fCO_2$, the annual flux was directed from the atmosphere to the sea.

P13L335: More appropriate and recent references are Touratier et al. (2016), Ingrosso et al. (2017), and Krasakopoulou et al. (2017), who estimated the anthropogenic CO2 in the three dense water formation area of the Mediterranean Sea.

We believe that the 2 references cited [*Schneider et al.*, 2010] and [*Palmiéri et al.*, 2015] give the relevant information in relation to the western basin of the Mediterranean Sea which is studied in our paper.

Technical comments  I suggest to improve the general quality of the figures.

This has not been done. The figures are the same.

P11L286: "P=0,0749" Substitute the coma with point.

This has been done.

References  Álvarez, M., Sanleón-Bartolomé, H., Tanhua, T., Mintrop, L., Luchetta, A., Cantoni, C., Schroeder, K., Civitarese, G., 2014. The CO2 system in the Mediterranean Sea: a basin wide perspective. Ocean Sci. 10, 69–92.  Carter, B. R., N. L. Williams, A. R. Gray, and R. A. Feely. 2016. Locally interpolated alkalinity regression for global alkalinity estimation. Limnol. Oceanogr. Methods 14: 268– 277. doi:10.1002/lom3.10087  Dickson, A.G., Sabine, C.L., Christian, J.R., 2007. Guide to best practices for ocean CO2 measurements. PICES Spec. Publ. 3, 191.  Ingrosso, G., Bensi, M., Cardin, V., Giani, M., 2017. Anthropogenic CO2 in a dense water formation area of the Mediterranean Sea. Deep-Sea

Research Part I 123, 118– 128. doi:10.1016/j.dsr.2017.04.004  Jordà, G., Schuckmann, Von, K., Josey, S.A., Caniaux, G., García-Lafuente, J., Sammartino, S., Ozsoy, E., Polcher, J., Notarstefano, G., Poulain, P.M., Adloff, F., Salat, J., Naranjo, C., Schroeder, K., Chiggiato, J., Sannino, G., Macías, D., 2017. The Mediterranean Sea heat and mass budgets: Estimates, uncertainties and perspectives. Progress in Oceanography 156, 174–208. doi:10.1016/j.pocean.2017.07.001  Krasakopoulou, E., Souvermezoglou, E., Giannoudi, L., Goyet, C., 2017. Carbonate system parameters and anthropogenic $CO_2$ in the North Aegean Sea during October 2013. Continental Shelf Research 1–13. doi:10.1016/j.csr.2017.04.002  Touratier, F., Goyet, C., Houpert, L., de Madron, X.D., Lefèvre, D., Stabholz, M., Guglielmi, V., 2016. Deep-Sea Research I. Deep-Sea Research Part I 113, 33–48. doi:10.1016/j.dsr.2016.04.003  Interactive comment on Biogeosciences Discuss., https://doi.org/10.5194/bg-2017-284, 2017.

**References**

Begovic , M., and C. Copin:Montegut (2002), Processes controlling annual variations in the partial pressure of $fCO_2$ in surface waters of the central northwestern Mediterranean sea (Dyfamed site), *Deep$Sea Research II*, *49*, 2031:2047. Huertas, I. E., A. F. Ríos, J. García:Lafuente, A. Makaoui, S. ` Rodríguez:Gálvez, A. Sánchez: Román, A. Orbi, J. Ruíz, and F. F. and Pérez (2009), Anthropogenic and natural $CO_2$ exchange through the Strait of Gibraltar, *Biogeosciences*, *6*, 647:662.

Millot (1999), Circulation in the Western Mediterranean Sea, *Journal of Marine Systems*, *20*, 423–442.   Palmiéri, J., J. C. Orr, J. C. Dutay, K. Béranger, A. Schneider, J. Beuvier, and S. Somot (2015), Simulated anthropogenic $CO_2$ storage and acidification of the Mediterranean Sea, *Biogeosciences, 12*(3), 781:802. Schneider, A., T. Tanhua, A. Körtzinger, and D. W. R. Wallace (2010), High anthropogenic carbon content in the eastern Mediterranean, *Journal of Geophysical Research, 115*(C12). Wanninkhof, R. (2014), Relationship between wind speed and gas exchange over the ocean revisited, *Limnology and Oceanography: Methods, 12*(6), 351:362.

References

Touratier, F., Guglielmi, V., Goyet, C., Prieur, L., Pujo-Pay, M., Conan, P., Falco, C., 2012. Distributions of the carbonate system properties, anthropogenic CO2, and acidification during the 2008 BOUM cruise (Mediterranean Sea). Biogeosci. Discuss. 9, 2709–2753. http://dx.doi.org/10.5194/bgd-9-2709-2012.

---

## Author Response (AR2)

**Major Comments**

My major reservation about this work is the difference between the measured fCO2 at the sea surface (fCO2sea) and the fCO2 derived from atmospheric xCO2 concentration (fCO2air). In 2013-2015 the sea surface mean annual fCO2 calculated at 18.25_C (the mean annual in situ temperature) was larger than the fCO2air derived from atmospheric data at the same temperature. This result is quite strange, because it means a CO2 outgassing from the sea surface to the atmosphere on annual average, which is in contrast with respect to the ongoing ocean acidification process and the general net anthropogenic CO2 uptake measured in the Mediterranean Sea by different research. In 2013-2015 I would expect an equilibrium between the fCO2sea
and fCO2air, or a slightly higher value in the fCO2air, as it was detected in the 1995-1997. How the authors can explain this issue?

In the 2 periods, 1995-1997 and 2013-2015, the CO2 annual flux is directed from the atmosphere to the sea in both cases, although the annual average of CO2 in surface seawater in 2013-2015 is higher than atmospheric fCO2. This is due to higher wind speed in autumn and winter when the surface water is undersaturated. This is well illustrated in the figure below for the time period 2013-2015. In the upper figure, the three thin lines indicate fCO2 atm.

This could be a good explanation, but it must be supported by a statistical analysis of the data. Is there a significant statistical difference in the wind speed between winter/spring/summer/autumn? From the figure proposed, the wind speed seems more or less the same during the different month.
On the figure below, we see that during the period May-September, the monthly values of the wind speed are almost 2 times lower than during the other months.

[Figure]

The mean annual CO2 flux is equal to -0.45 mol .m$^{-2}$.yr$^{-1}$ using the exchange coefficient of [Wanninkhof, 2014].

How was calculated the mean annual CO2 flux? If this is the average of the daily CO2 flux, it is also necessary to report the standard deviation. Please clarify.

It is calculated as the mean of individual hourly values of the product of the gas exchange coefficient and the atmosphere-sea $fCO_2$ gradient. We are not sure to understand why the reviewer asks for the standard deviation which is necessary large because of the seasonality of both terms, the wind speed and $fCO_2$ at the sea surface.

They suggested the contribution of the Atlantic Ocean as a source of anthropogenic carbon, but I do not understand how the Atlantic surface waters can be relatively enriched in anthropogenic carbon.

[Huertas et al., 2009] conducted a sampling program at eight fixed stations in the Strait of Gibraltar to study natural and anthropogenic carbon exchange between the Atlantic Ocean and the Mediterranean Sea. Their results show that Atlantic water has a higher concentration of anthropogenic carbon than Mediterranean water. A decreasing vertical gradient of Cant in the water column is observed, the upper layers being enriched in Cant (Figures 5 and 6).

My doubts remain. Since Cant cannot be measured directly, as it cannot be chemically discriminated from the bulk of dissolved inorganic carbon, different approaches for its indirect estimation have been developed. All the proposed approach do not give good results in the surface layer, due to the effect of the biological activity and the strong dynamic of this portion of the water column. For these reasons usually the the surface waters (0-200m) is not considered in the estimation of Cant. Touratier et al. (2012) strongly criticized Huertas et al., 2009 to calculate the Cant in the surface layer, and Palmieri et al. (2015) also reported Cant calculation of the surface layer. So, is the Atlantic Ocean a sink or a source of Can respect to the Mediterranean Sea? At the moment we do not have clear scientific evidence to answer at this question.

We added in 4.2.2. the conclusion of the paper by Flecha et al. [2012]. In this study, 3 observational methods using 3 different back calculation techniques for the $C_{ant}$ concentration assessment were used to calculate the anthropogenic carbon inventory in the Gulf of Cadiz. The authors also conclude that there is a net import of $C_{ant}$ from the Atlantic towards the Mediterranean Sea. We have written lines 371-378:
« The concentration of oceanic anthropogenic carbon, $C_{ant}$, is not a directly measurable quantity. To estimate it, several empirical methods have been developed. Flecha et al.[2012] computed the anthropogenic carbon inventory in the Gulf of Cadiz. They used observations made during a cruise in October 2008 throughout the oceanic area covered by the Gulf of Cadiz and the Strait of Gibraltar to estimate $C_{ant}$ with 3 methods: $\Delta C^*$ [*Gruber et al.*, 1996] ,TrOCA [*F Touratier and Goyet*, 2004; *F. Touratier et al.*, 2007] , $\varphi C_T^0$ [*Vazquez-Rodriguez et al.*, 2009]. In the 3 cases, their results indicate a net import of $C_{ant}$ from the Atlantic towards the Mediterranean through Gibraltar. »

Moreover, this is in contrast with the end of the discussion where the authors say that (P13L331) "The Mediterranean Sea is actually able to absorb more anthropogenic CO2 per unit area".
As stated in the text, surface waters of the Mediterranean basin have a relatively low Revelle factor, close to 10, due to a high alkalinity and a high temperature and therefore have a relatively high uptake capacity for Cant.

The answer is not pertinent to may question. I try to be more clear. How the Atlantic Ocean can be a source of the Can if (as the authors say P13L331) "The Mediterranean Sea is actually able to absorb more anthropogenic CO2 per unit area"?

In the strait of Gibraltar, Atlantic waters flow eastward to the Mediterranean Sea located in the upper layers of the water column while the westward Mediterranean outflow occupies the deeper part. In the shallower depth, the Atlantic waters are enriched in anthropogenic $CO_2$ as they have been recently in contact with the atmosphere while the deep Mediterranean waters have not been in interaction with the atmosphere since a long time period.

Maybe there are other causes which could explain the fCO2 increase at the sea surface observed in 2013-2015, such as a stronger and deeper winter vertical mixing with CO2 enriched LIW.

The reviewer is right. A strong interannual variability of winter convection events between the two studied periods has been observed and must be taken into account to interpret the total temporal change of the computed increase of DIC. This is detailed in paragraph 4.3, lines 323 -329.

In this revised version of the manuscript, we expanded the discussion on the potential effect of natural variability. We wrote, lines 348-369:
"**4.2.1** Natural variability

Time series of mixed layer depth, MLD, show a strong variability in winter at interannual scale. During the two periods, 1995-1997 and 2013-2015, the winter MLD never exceeded 220 m, whereas values over 300 m were observed in 1999 and especially in February and March 2006 with values close to 2000 m [Coppola et al., 2016; Pasqueron de Fommervault et al., 2015]. These episodes of strong and deep vertical mixing must have entrained DIC rich LIW in the surface waters. This could be causing an increase in DIC between the 1995-1997 and 2013-2015 periods. Monthly surface samples collected at the Dyfamed time series station between 1998 and 2013 indicate an increasing DIC trend of 1.35 μmol kg$^{-1}$ yr$^{-1}$. This value is known with great uncertainty ($r^2 = 0.05$) because of the large seasonal variability displayed in the monthly samples [*Gemayel et al.*, 2015]. Nevertheless, this value is closer to the trend we calculated between the two periods, 1993-1995 and 2013-2015 (1.40 µmol kg$^{-1}$ yr$^{-1}$) than to the trend inferred from the atmospheric increase (1.15 µmol kg$^{-1}$ yr$^{-1}$). On DYFAMED time series, we find no evidence that the strong increase in MLD observed during winters 1999 and especially 2006 resulted in a further increase in DIC.

The monthly cruises of the Dyfamed time-series study have also been analyzed in order to investigate the hydrological changes and some biological consequences over the period 1995-2007 [*Marty and Chiavérini*, 2010]. These authors show that extreme convective mixing events such as recorded in 1999 and 2006 are responsible of large increases in nutrient content in surface layers and conclude that the biological productivity is increasing especially during the 2003-2006 period, which could lead to a larger consumption of carbon, i.e. a decrease of DIC. "

Finally, additional information about the water mass exchange throughout the Strait of Gibraltar and its temporal variation are needed.

This is analyzed and discussed in [Huertas et al., 2009], see for instance figure 7. See also [Schneider et al., 2010], table 2.

These can be found in the recent review of Jordà et al. (2017) which may provide more insights for this work. The authors found a DIC increase larger than expected from equilibrium with atmospheric CO2. They hypnotized a _15% contribution of the Atlantic Ocean as a source of anthropogenic carbon to the Mediterranean Sea through the strait of Gibraltar. I think that the analysis presented in the manuscript are not sufficient to support such hypothesis and the authors should provide a lot more analysis and discussions.

This is detailed in the paragraph 4.3.

Why the author do not consider the recent review of Jordà et al. (2017) about the water mass exchange in the Strait of Gibraltar?

In their article, Jorda et al review various estimates and uncertainties regarding the heat and mass flow across the Strait of Gibraltar. They update and analyze data from existing literature and indicate possible directions for improving methodological and observational methods to estimate the heat and mass content of the Mediterranean Sea. We do not think it is necessary to add this reference to our article

Moreover, the Mediterranean Sea overturning circulation and the sites of dense water formation could play a very important role in the sequestration of anthropogenic CO2 and in the ocean acidification of the Mediterranean Sea.I think that the authors should read the recent papers of Touratier et al. (2016), Ingrosso et al. (2017), and Krasakopoulou et al. (2017), who estimated the anthropogenic CO2 in the Gulf of Lion, Adriatic Sea, and the Aegean Sea respectively.

Certainly the reasons why the Mediterranean Sea water column stores large amounts of anthropogenic CO2 are due to the fast deep water formation processes combined with surface water having high potential to take up Cant due to a relatively low Revelle factor.

Ok, but why the author do not want to consider and to cite these recent papers which estimate the Cant in Mediterranean Sea? Touratier et al. (2016) also estimate the Cant in an area very close respect to the DYFAMED site.

Cant estimates in the Mediterranean Sea differ greatly from one method to another. Some critics of the TROCA back calculation technique are put forward more specifically for its application in the Mediterranean Sea. We do not want to enter this debate (Yool et al, 2010, Schneider et al., 2010, Palmieri et al., 2015).

The authors try to assess the influence of physical and biological process on the seasonal and inter-annual variation of fCO2. To do this, they used a simple analysis of the change of fCO2@13 (fCO2 normalized to the constant temperature of 13_C) as a function of SST, which is not sufficient to achieve the scope. I suggest to quantify (1) the air-sea CO2 exchange and (2) the thermal/not-thermal contributions on the fCO2 variation with the method of Takahashi et al. (2002). In this way the authors could clarify how fCO2 seasonal variation is affected by physical (i.e. temperature, mixing, and air-sea CO2 exchange) and biological processes (i.e. photosynthesis, respiration, and calcification).

The objective of our paper is to compare the time change of surface fCO2 measurements made at 2 very close locations, Dyfamed and Boussole, at an interval of 18 years. The processes that govern the distribution of fCO2 at the annual scale at the same site have been analyzed in detail in a publication entitled "Processes controlling annual variations in the partial pressure of CO 2 in surface waters of the central northwestern Mediterranean Sea (Dyfamed site)[Begovic and Copin-Montegut, 2002]. For instance, the figure 8 in this paper is a good illustration of the relative importance of individual processes which govern the distribution of DIC over an annual cycle. For this reason, we decided not to repeat this well-argued description which is already published.

Specific Comments

P4L93: If the authors followed the standard operational procedures, the reference of Dickson et al. (2007) could be added to Edmond (1970).

The reference to Edmond (1970) is line 102.

Where is the reference of Dickson et al. (2007)? Did the authors follow the standard operational procedures?

The standard  procedures have been applied . We have added the reference Dickson et al. (2007).

P5L126: I propose to consider here the the method of Takahashi et al. (2002) and to present the temporal variation of the thermal and not-thermal fCO2 as differences (dfCO2) with respect to the February, chosen as reference month because it usually presents the lowest temperature and the minimum biological activity.

We have chosen to estimate the difference between the values of the thermal component fCO2@13 two decades apart according to the temperature (14 temperature steps of 1°) and not to the time. This approach is more quantitative than a comparison of monthly values because we know that key processes which control the fCO2@13 distribution such as the beginning of the bloom depend more directly on a narrow temperature threshold (13-14 °) while it may vary up to one month.

P5L128: The "remineralization" is a biological activity. Please modify/clarify the sentence.

This has been done (line 139).

P5L130: Do the authors have oxygen data? The examination of the O2/DIC or AOU (apparent oxygen utilization)/DIC ratio would provide useful information about the influence of biological activity to the observed fCO2 variation. Also satellite data of Chloro-Phyll phyll a concentration may help, which nowadays are easy to get

See our comment above before Specific Comments.

Do the authors have oxygen data? I do not found answer to this question.

We do not have oxygen data.

P6L134: "The contribution of air-sea exchange is not significant". In order to support this sentence, please can the authors calculate the air-sea CO2 flux and estimate the real influence of this process?

This has been done, lines 146-148.

P6L150: Levantine Intermediate Water (LIW) originates in the Eastern Mediterranean and takes years to reach the Ligurian Sea. Due to the organic matter remineralization processes, the LIW presents low dissolved oxygen concentration and high CO2 levels (Álvarez et al., 2014), even higher than then the atmospheric levels. Taking into account these considerations, in the present study, the increase of total dissolved inorganic carbon observed in 2013-2015 can be related to a stronger and deeper winter vertical mixing with CO2 enriched LIW?

The reviewer is right. A strong interannual variability of winter convection events between the two studied periods has been observed and must be taken into account to interpret the total temporal change of the computed increase of DIC. This is detailed in paragraph 4.3, lines 323 -329.

As reported by Alvarez et al. (2014), the LIW during its westward flows can increase DIC and lower pHT of different Mediterranean basin. P7L197: "mixing with enriched deep waters" please substitute with "mixing with CO2- enriched deep waters". This may support the hypothesis of a general DIC increase generated by mixing with LIW, but further analysis and more discussions are needed.

No reply to this comment

In our revised manuscript, we added a paragraph (4.2.1 Natural variability, lines 348-369) to evaluate the impact of natural variability as an increase in vertical mixing that could have entrained a larger mixing with LIW enriched in DIC. We noted that, with existing available data in the literature, we found no significant signature of an increase in DIC between the 2 time periods, 1995-1997 and 2013-2015.

P8L199: During summer, due to the high sea surface temperature, the CO2 flux from the sea to the atmosphere could also play an important role. Please consider also this process in addition to the biological drawdown of carbon.

See our comment above before Specific Comments

I do not understand why the author do not consider the influence of the CO2 flux from the sea to the atmosphere.

Our computations consider the budget of annual DIC in the mixed layer. They are independent of the seasonal variation of the flux.

P9L223: "Changes of seawater carbonate chemistry in surface waters". This section needs some modification/clarification. L223-227 seems more appropriate for the Material and methods.

In Material and methods, we consider the DIC and Alk analysis of the
seawater samples taken at Boussole during the servicing cruises to the
mooring. In the section 3.4, we consider the derived values of DIC and pH from the analysis of the 2 time series of fCO2.

L229-234: DIC and pH are derived parameters. They are calculated from total alkalinity and fCO2. Due to this reason, the fCO2-DIC and fCO2-pH may not have sense and the near perfect R2 is not significant. Please, can the authors clarify this issue?

This has been changed. We just compute DIC and pH as suggested.

P9L229: pHT refers to the pH on the total scale. But the authors calculated the pH on the seawater scale (P9L228) which is conventionally denoted as pHsws. Please substitute in all the manuscript/figures the pHT with pHsws.

We compute pH on the seawater scale. We delete T .We indicate in the text that the change of pH is computed at the mean in situ temperature 18.25°C

You should substitute in all the manuscript/figures the pHT with pHsws. Not only delete T. Only pH is not correct and ambiguous.

We use pH $_{sws}$ in the manuscript

P11L259: Any references which can support that Atlantic surface waters are relatively enriched in anthropogenic carbon and why?

See [Huertas et al., 2009].

In the same paper the TrOCA approach measured a greater Cant in the Mediterranean waters.

See our previous comment p.6

Even if the Atlantic surface water could be enriched in CO2, I do not think that it could preserve this property. An air-sea equilibrium, mixing, and biological processes may happen during the long time that Atlantic surface water spent to reach the Ligurian Sea from the Gibraltar Strait.

The depth of the surface water layer of the Atlantic entering the Mediterranean Sea through the Strait of Gibraltar is close to 200 meters. It would take a few months to reach the

Dyfamed zone assuming a lower estimate of the average current close to 10 cm / s on its route along the Algerian coast and then northwards [Millot, 1999]. This indicates that CO2-enriched Atlantic water may retain its signature during this relatively short
period of time.

P11L270-272: More discussion and references are needed to support this sentence.

This was not correct. As indicated earlier, and illustrated in the figure, although the annual average of fCO2 in surface seawater was higher than atmospheric fCO2, the annual flux was directed from the atmosphere to the sea.

P13L335: More appropriate and recent references are Touratier et al. (2016), Ingrosso et al. (2017), and Krasakopoulou et al. (2017), who estimated the anthropogenic CO2 in the three dense water formation area of the Mediterranean Sea.

We believe that the 2 references cited [Schneider et al., 2010] and [Palmiéri et al., 2015] give the relevant information in relation to the western basin of the Mediterranean Sea which is studied in our paper.

Technical comments I suggest to improve the general quality of the figures.

This has not been done. The figures are the same.

We reworked the figures. We hope it will be more satisfying.

P11L286: "P=0,0749" Substitute the coma with point.

This has been done.

We do not quote this paper as it  has not been accepted for publication in Biogeosciences and the reviews available in BGD are severe.

Second interactive comment on "Increase of dissolved inorganic carbon and decrease of pH in near surface waters of the Mediterranean Sea during the past two decades" by Liliane Merlivat et al.

Anonymous Referee #2

MAJOR CONCERNS:

In my previous review of the submitted manuscript from Merlivat et al., I mentioned two general concerns that led me to recommend that in-depth revisions were needed. First, uncertainties appeared underestimated and poorly described, and second natural variablity was ignored as a possible explanation for some of the change seen between the two 3-year periods.

In the revised manuscript, the authors have tried to clarify their text in regards to my first concern. Yet the discussion of these results, appears unclear, imprecise, and does not offer a clear statistical demonstration that the differences between the two time periods are significant. More detailed concerns about the generally poor description of uncertainty analyses are provided below in the comments concerning lines 108-109, 241-246, 254-255, and 275-276.

We bring details under these comments. We have modified the section 4 "Discussion" which is now organized as follows:

4.1 Time change of surface alkalinity

4.2 Drivers of the temporal change of DIC in surface waters

4.2.1 Natural variability

4.2.2 Anthropogenic carbon exchange through the Strait of Gibraltar

4.3 Long term trends in surface DIC and pH

As for my second major concern, the authors response is unsatisfactory. Although this concern is mentioned briefly in the Introduction, the authors have just cut and paste an entire sentence from my Review, word for word. The same concern is mentioned briefly in the Abstract, Discussion, and Conclusion. Much more text is devoted to the explanation of anthropogenic change rather natural decadal variability.

We have added more details in the paragraph 4.2.1 " Natural variability"

Overall, I am disappointed with the authors responses to both of my previous major concerns. Substantial improvements would still be needed to clarify these points to a satisfactory level before I could recommend that the manuscript would be publishable.

DETAILS:

lines 51-53: Change
 The quantitative estimation of anthropogenic CO2 storage in the
Mediterranean Sea based on experimental data is very inaccurate, of the order of a factor two [Huertas et al., 2009; Touratier and Goyet2009] " to
"Estimates of anthropogenic storage in the in the Mediterranean Sea differ by about a factor of two [Huertas et al., 2009; Touratier and Goyet, 2009] "

This has been done.

lines 56-57: "[McKinley et al, 2011]" is repeated 5 times. This sentence is plagiarized from my review.

This has been corrected.

lines 58-59: Suggest changing of time is a way to detect a possible trend in DIC." to "iIs useful to assess trends and variability of DIC."

This has been corrected.

line 62: change "very close" to "nearby" or "adjacent".

This has been done.

line 86: delete "depth"

This has been done.

line 96: Incorrect format in sentence for citation.

This has been corrected.

line 101: change "meters" to "m"

This has been changed.

lines 108-109: Errors on fCO2 calculated from DIC and Alk, each having uncertainties of about 3 umol/kg, is about 5% based on results from Dickson (2010, Table 1.6). For a base value of fCO2=400 uatm, that would imply that the uncertainty in calculated fCO2 from Merlivat et al. is +/- 20 uatm. This is four times larger than the estimate quoted by the authors based on the paper from Millero et al. (2007). The authors need to mention this more recent study and the much higher uncertainty in calculated fCO2 that is implied from that.

Table 1.6 of Dickson (Acidification_Handbook_EU.,2010) refers to overall uncertainty on measurements performed using various techniques and coming from uncertainties on dissociation constants. As stated on page 37 the marine chemistry community rarely considers such combined uncertainty, because marine scientists usually consider measurements performed using a single technique, a single set of equilibrium constants, and are then interested in a precision estimate. This is our case, as measurements performed during the two time periods were performed using the same instrument type and the processing was made using the same dissociation constants. If a systematic error occurs, it will not significantly affect the changes estimated between the two time periods. Hence, we keep the error estimate according to (Millero 2007). It is also important to notice that in a recent study, Alvarez et al. (2014) recommend to use the dissociation constants of Mehrbach et al. (1973) refitted by Dickson and Millero(1987) in the Mediteranean Sea.

We have not added the reference to Dickson (2010) because of the reasons explained above. We have added a reference to Alvarez et al. (2014).

line 138: change this last effect should be negligible" to "the effect of nutrients may be neglected".

This has been corrected.

line 150: change "meters" to "m".

This has been changed.

line 236: delete It is interesting to note that

This has been corrected.

Table 1: Over the final 2 columns it is marked "Temporal trend". Actually this is not a trend (change per unit time) but just the difference between the 2 periods.

We have written "Temporal change".

lines 241-246: Confusing explanation of the uncertainties in the estimates of the difference dfCO213. just because there are 3 CARIOCA sensors and each is supposed to have an accuracy of +/-2 uatm does not mean that the accuracy of the estimate (presumably the time average) for each period is 2/(3)^0.5. The standard deviation of the measurements in each time period is much larger (7 to 28 uatm). The standard deviation of the difference is much larger still. The authors are not clear about what they are referring to exactly when they say that the accuracy on the difference is estimated to be 1.6 uatm." They may be referring to the standard error of the mean, but they do not say so explicitly. In any case, judging from the numbers in Table 1, the last 2 columns suggest that the difference between the 2 time periods is not always even significant. The authors would need to provide a significance test to show that the means of the two time periods actually differ significantly.

We have modified and rewritten this part as follows:
"We have estimated the uncertainties in the estimates of the difference $dfCO_2@13$ with 2 methods. Firstly, the arithmetic mean of $dfCO_2@13$ is equal to 33.17µatm, with a standard deviation, SD, and standard error, SE, respectively equal to 6.29 µatm and 1.68 µatm. A 95% confidence interval is thereby achieved within 1.96 SE, i.e 3.29 µatm. A second approach consists of computing a weighted average of the mean of $dfCO_2@13$. In this case, mean weighted value of $dfCO_2@13$ over the whole range of temperature is estimated, the weights being equal to the variance of $dfCO_2@13$ in each temperature step. It is equal to 32.70 µatm. The weighted SD, and the associated SE, of the 14 data points are respectively equal to 4.85 µatm and 1.30 µatm. A 95% confidence interval is achieved within 2.54 µatm. The difference between the two mean $dfCO_2@13$ estimates is 0.47 µatm, well below SE. In the following, we have chosen the former method which produces a more conservative estimate."
We hope that it is clearer.

lines 254-255: The authors do not provide enough detail on how they made their calculations for the changes in DIC and pH and the corresponding uncertainties. Are they using mean values of fCO213 and their for the two time periods? Are they using the standard error of the mean for the uncertainty? Without substantially more detail, I am left with the impression that they are underestimating the uncertainties. The uncertainties they do provide for the differences in DIC and pH are much smaller than the measurement uncertainties.

We indicate how we made the calculations for the change in DIC and pH. We have written (lines 281-285):

"The error on $dfCO2@13$ ,+/-3.3µatm, has been propagated to compute the uncertainty on dDIC and $dpH_{SWS}$. This makes the implicit assumption that there is no systematic error on DIC and $pH_{SWS}$ derived from $fCO2@13$ between the two time periods; in particular, mean temperature and salinity remain the same (section 3.2). This is further discussed in section 4.1."

lines 275-276: change the corresponding amount of anthropogenic carbon taken up from the atmosphere in order to maintain a chemical equilibrium at the sea surface would be equal"
to
 the corresponding change in surface DIC, assuming air-sea equiliibrium, would be would be

 This has been changed

It is unsure that these annual mean calculations are adequate since it has been shown previously that the air-sea flux of anthropogenic carbon varies seasonally. This doubt is further supported by the authors own statements in their lines 291-297.

Our computations consider the annual budget of DIC in the mixed layer. They are independent of the seasonal variation of the air-sea flux of anthropogenic $CO_2$.

line 332 : improper format for citation line 333 : improper format for citation line 339 : improper format for citation line 345: improper format for both citations.

We have made the corrections in the 3 cases.

line 355: The authors still use the symbol "pH_T", unlike what they say in their response to my previous comments.

This has been corrected.

line 356: improper format for citation.

line 360: improper format for citation.

line 364: improper format for citation.

We have made the corrections in the 4 cases.

Section 4.4: The discussion of changes in pH is inadequate. It does not consider that the change in pH depends not only on the change in line 354-370: Section 4.4

This subsection appears particularly weak. It is descriptive but offers no real discussion. Why is the magnitude of the change of pH in the Mediterranean Sea more than the global ocean average. Is that difference significant? Why is it more than in the Iceland Sea but less than in the Irminger Sea. The authors have made no effort to discuss the causes of these differences. The last sentence only talks about the change in  anthropogenic carbon, but that is not so clearly related to the change in pH since the alkalinity in the Mediterranean Sea is higher than at the other sites that are mentioned. Thus anthropogenic DIC increase there should be higher there although the pH change may be similar to that for the global ocean (Palmieri et al., 2015). But changes in pH are confusing because of the log scale. An absolute change in pH actually represents a relative change in H+.

This subsection has been deeply modified. The magnitude of the change of pH in the Mediterranean Sea does not differ significantly from the global ocean average. This is now clearly written. We have modified the title of the section "The signal of acidification" into " 
[revised manuscript text omitted]

---

## Author Response (AR3)

In their second revision, the authors have addressed my two major concerns: (1) apparently underestimated uncertainties, and (2) the need to discuss natural variability as a possible explanation of part of the change seen between the two 3-year periods two decades apart. My recommendation would then be to publish this study after my remaining minor concerns, listed below, have been addressed. Most but not all of these concerns relate to trying to improve the English.

ABSTRACT

Page 1, line 20: change "Two three-year-long time series" to "Two 3-year time series".

This has been done.

1, 21: change "10m" to "10 m"

This has been done.

1, 31-32: The last sentence of the abstract is not clear.

It has been changed

INTRODUCTION

2, 41-42: The three groups of citations in brackets with semicolons in between each group is strange. It does not follow BG conventions.

It has been corrected.

2, lines 47 and 49: Citations are given with incorrect formatting.

It has been corrected.

2, 55: change "after" to "only after"

3, 68: change "18 years" to "18-year"

4, 93: change "installed" to "attached"

We have made the corrections in the 3 cases above.

4, 98-103: accuracy vs. precision may need revisiting

We think this is discussed in Hood and Merlivat, 2001, page 116.

4, 113: The authors need to be more specific about what they mean by "Error on $fCO_2$". I think they mean the propagated "uncertainty" (not the "error") based only on their previous uncertainty estimates for measurements of ALK and DIC, neglecting uncertainties in the

equlibrium constants. This should be spelled out because it makes a big difference in the propagated uncertainty estimate. The authors should not use the word 'error', as detailed further below.

We have written uncertainty in place of error.

5, 132-137: Fig. 2 caption:

- The caption does not mention where the data were collected.

- "(d), (e), (f), seasonal variability" is unclear and not a sentence.

The Fig.2 caption has been rewritten

6, 142: change "extrema" to "extremes"

It has been corrected.

6, 147: change "thermodynamic" to "temperature"

It has been corrected.

6, 150: citation has incorrect format.

It has been corrected.

6, 153: "Biology accounts for" is vague.

We have written: “Biological processes account for the decline…”

6, 160: change "CO2 air-sea flux" to "air-sea CO2 flux"

This has been changed.

7, 164 and 167: Do the authors mean "internal waves" rather than "intertial waves"?

We mean “inertial waves”. Their period, computed with hourly measurements of temperature at 10 m between July and October 2014 is 17.4 H.

.7, 169: change "leading" to "corresponding"

7, 172: change "over" to "during"

7, 199: add "s" to "temperature"

8, 204: delete "very"

We have made the corrections in the 4 cases above.

8, 209 and 211: The authors need to inform readers what the uncertainties in salinity are supposed to represent (std. deviation, standard error of the mean, or something else).

The paragraph has been rewritten.

8, 224: add a comma after "winter"

This has been corrected.

8, 226: change "is clearly highlighted for the whole range of temperature." to "is evident across the range of temperatures."

This has been changed.

9, Fig. 3 caption:

- yellow dots are mentioned in the figure caption for panel (a) but none are visible in the actual figure.

The yellow dots have been enlarged.

10, 249-258: The authors have confused me entirely here. In English, paragraphs should be indicated either by an indentation or a blank line as a separation before the start of a new paragraph. The authors have done neither. Nonetheless, I was usually able to guess when there is a new paragraph in their manuscript, i..e., when the previous line does not extend all the way to the right margin. However, this sloppy formatting makes it impossibile to tell what

is going on in lines 24ç-258 and perhaps beyond. There appear to be two 1-sentence pseudo-paragraphs in the beginning, followed by a 3-sentence paragraph. But maybe all of this is intended to be part of the Table 3 title? In any case, with the current structure, one cannot tell where the Table title ends and where the ensuing text in the manuscript begins. All very confusing!

We have reorganized the paragraph 3.3.2 and the table 1, lines 239-287. We hope it is clearer now.

10, line 259: What "mean" is being discussed here?

(now, line 248). A correction has been made.

11, line 273: If the two methods give results that are not significantly different, i don't think that one can justify using one vs. the other because "the former method produces a more conservative estimate." Furthermore, just because the value is lower does not mean it is "more conservative".

(now, line  268). We have deleted "which produces a more conservative estimate".

11, 281: It is incorrect to use the word "error" here. The 'error' cannot be known because one cannot know the true value. The 'error' also has a sign; it cannot be reported as +/-x (unlike the uncertainty). Thus "error on" should be changed to "standard uncertainty of". Later in the same line, "uncertainty" should be changed to "combined uncertainty in".

(now, lines 294-295). We have made corrections and write: "The uncertainty of dfCO2@13, 3.3µatm, has been propagated to compute the combined uncertainty in dDIC and $dpH_{SWS}$. »

11, 282-284: It should also be stated that uncertainties in the equlibrium constatnts are neglected in this propagation of uncertainties.

(now , lines 295-296). This has been indicated.

11, 285-287:

- units for DIC are given strangely (remove the spaces before "mol"

- pH has no units. Remove "unit" in lines 286-287.

This has been corrected.

What is meant is "the propagated uncertainty accounting only for standard uncertainties of the measurements and ignoring uncertainties in the constants".

We agree.

12, 298: Problems with formatting of Table title and foonotes

We have reorganized the presentation of the table titles and the footnotes.

DISCUSSION

13, 344: The authors say that "The difference between these two values is significant." Such statements should come with the name of the specific statistical test used and the resulting p value that allows the author to make such a statement.

(now lines 348-352). The sentence has been modified.

The uncertainty in DIC resulting from the change of sea surface or atmospheric $fCO_2$ is the result of propagating uncertainty on changes of fCO2 known within a 95% confidence interval (lines 260 and 318).

13, 349: add "the" before "mixed layer depth".

13, 353: add hyphen between 'DIC' and 'rich'.

14, 354: 'This' what? Never use 'This' by itself at the beginning of a sentence. In this case, the authors could say 'This entrainment'.

14, 366: change 'of' to 'for'

We have made the corrections in the 4 cases above.

15, 388: 'in excess of 17+/-10%' is unclear. Do you mean it could be more than 27%?

The sentence has been changed. We mean (17+/-10)%.

15, 396: delete 'would'

15, 400: add a hypen between 'Long' and 'term'

15, 419: delete 'have also to be taken into account'. That phrase is unnecessary and messes up the sentence.

15, 420: delte 'rather'

16, 424: delete 'absorption'

16, 428-429: change 'Mediterranean anthropogenic acidification' to 'anthropogenic acidification of the Mediterranean Sea'

16, 430: change add 'of the Mediterranean Sea' after 'pH'

16, 431: add 'that of' after 'from'

The corrections have been made in the 8 cases above.

16, 436: 'considerable short-time' is an oxymoron. The meaning is unclear.

(now, line 440). The sentence has been modified.

16, 442: change 'as a source' to 'a substantial amount'

( now, line 447). This has been changed.

16, 443-444:

- change 'towards' to 'to'

- delete '('

- rather than saying ", close to 10% ([Schneider et al., 2010] or 25% [Palmiéri et al., 2015]", the authors should provide their estimate and follow that with something like "which lies between estimates of 10% by Schneider et al. [2010] and 25% by Palmieri et al. [2015].

( now line 448-449).Changes have been made.The sentence has been modified.

16, 451: the '2' shoud not be subscripted in CO2SYS.

This has been corrected.

REFERENCES

The references are hard to read. Please add a line space between them or provide them each with hanging indentation at the beginning.

This has been corrected.

[revised manuscript text omitted]

Liliane Merlivat 31/8/18 12:19
Liliane Merlivat 31/8/18 12:19
Liliane Merlivat 31/8/18 12:21
Liliane Merlivat 31/8/18 12:21
Liliane Merlivat 31/8/18 12:41
Liliane Merlivat 31/8/18 12:41
Liliane Merlivat 31/8/18 12:45
Liliane Merlivat 29/8/18 15:43
Liliane Merlivat 29/8/18 15:42
Liliane Merlivat 29/8/18 15:43
Liliane Merlivat 29/8/18 15:44
Liliane Merlivat 29/8/18 15:44
Liliane Merlivat 31/8/18 12:20
Liliane Merlivat 30/8/18 17:07
Liliane Merlivat 30/8/18 17:07
Liliane Merlivat 30/8/18 17:11
Liliane Merlivat 30/8/18 17:07
Liliane Merlivat 30/8/18 17:07
Liliane Merlivat 30/8/18 17:08
Liliane Merlivat 30/8/18 17:07

[revised manuscript text omitted]

Liliane Merlivat 3/9/18 12:24

Unknown

Liliane Merlivat 31/8/18 12:59